# Structural basis for cytoplasmic dynein-1 regulation by Lis1

John P Gillies[1†], Janice M Reimer[1†], Eva P Karasmanis[1†], Indrajit Lahiri[1,2], Zaw Min Htet[1], Andres E Leschziner[1,3]*, Samara L Reck-Peterson[1,4,5]*

[1]Department of Cellular and Molecular Medicine, University of California, San Diego, San Diego, United States; [2]Department of Biological Sciences, Indian Institute of Science Education and Research Mohali, Mohali, India; [3]Division of Biological Sciences, Molecular Biology Section, University of California, San Diego, San Diego, United States; [4]Division of Biological Sciences, Cell and Developmental Biology Section, University of California, San Diego, San Diego, United States; [5]Howard Hughes Medical Institute, Chevy Chase, United States

**Abstract** The lissencephaly 1 gene, *LIS1*, is mutated in patients with the neurodevelopmental disease lissencephaly. The Lis1 protein is conserved from fungi to mammals and is a key regulator of cytoplasmic dynein-1, the major minus-end-directed microtubule motor in many eukaryotes. Lis1 is the only dynein regulator known to bind directly to dynein's motor domain, and by doing so alters dynein's mechanochemistry. Lis1 is required for the formation of fully active dynein complexes, which also contain essential cofactors: dynactin and an activating adaptor. Here, we report the first high-resolution structure of the yeast dynein–Lis1 complex. Our 3.1 Å structure reveals, in molecular detail, the major contacts between dynein and Lis1 and between Lis1's ß-propellers. Structure-guided mutations in Lis1 and dynein show that these contacts are required for Lis1's ability to form fully active human dynein complexes and to regulate yeast dynein's mechanochemistry and in vivo function.

**\*For correspondence:**
aleschziner@ucsd.edu (AEL);
sreckpeterson@ucsd.edu (SLR-P)

[†]These authors contributed equally to this work

## Editor's evaluation

This manuscript reports the first high-resolution (3.1Å) structure of a dynein–Lis1 complex. Guided by their cryo-EM structure, the authors make mutations to show that the two Lis1 binding sites (ring and stalk) on the dynein motor are important for both dynein's in vivo function in *S. cerevisiae* and for the formation of human dynein/dynactin complexes.

## Introduction

Cytoplasmic dynein-1 (dynein) is a minus-end-directed microtubule motor that is conserved in many eukaryotes. Mutations in dynein or its regulators cause a range of neurodevelopmental and neurodegenerative diseases in humans (*Lipka et al., 2013*). In metazoans and some fungi, dynein transports organelles, RNAs, and protein complexes (*Reck-Peterson et al., 2018*). Dynein also positions nuclei and centrosomes and has a role in organizing the mitotic spindle and in the spindle assembly checkpoint (*Raaijmakers and Medema, 2014*). Given the essential nature of dynein in many organisms, the yeast *Saccharomyces cerevisiae* has been an important model system for studying dynein's mechanism and function as dynein and its regulators are conserved, but nonessential in this organism. In yeast, deletion of dynein or dynein's regulators results in defects in positioning the mitotic spindle (*Eshel et al., 1993*; *Kormanec et al., 1991*; *Lee et al., 2003*; *Li et al., 2005*; *Li et al., 1993*; *Muhua et al., 1994*; *Sheeman et al., 2003*).

Both dynein and its regulation are complex. Cytoplasmic dynein is a dimer of two motor-containing AAA+ (**A**TPase **a**ssociated with various cellular **a**ctivities) subunits, with additional subunits all present in two copies. In the absence of regulatory factors, dynein exists largely in an autoinhibited 'Phi' conformation (*Torisawa et al., 2014*; *Zhang et al., 2017*). Active moving dynein complexes contain the dynactin complex and a coiled-coil-containing activating adaptor. We will refer to these complexes here as 'active dynein' or 'dynein–dynactin–activating adaptor complexes' (*McKenney et al., 2014*; *Schlager et al., 2014*; *Trokter et al., 2012*; *Figure 1A*). Some of these activated dynein complexes contain two dynein dimers, and they move at a faster velocity than complexes containing a single dynein dimer (*Elshenawy et al., 2020*; *Grotjahn et al., 2018*; *Htet et al., 2020*; *Urnavicius et al., 2018*). About a dozen activating adaptors have been described in mammals, including members of the Hook, Bicaudal D (BicD), and Ninein families (*Canty and Yildiz, 2020*; *Olenick and Holzbaur, 2019*; *Reck-Peterson et al., 2018*). In yeast, the coiled-coil-containing protein Num1 is required to activate dynein in vivo (*Heil-Chapdelaine et al., 2000*; *Lammers and Markus, 2015*; *Lee et al., 2003*; *Sheeman et al., 2003*) and is likely a dynein-activating adaptor.

In addition to dynactin and an activating adaptor, dynein function in vivo also requires Lis1. Genetically, Lis1 is a positive regulator of dynein function (*Geiser et al., 1997*; *Liu et al., 1999*; *Xiang et al., 1995*) and is required for most, if not all, dynein functions in many organisms (*Markus et al., 2020*). In humans, mutations in both Lis1 (*PAFAH1B1;* hereafter called Lis1) and the dynein heavy chain (*DYNC1H1*) cause the neurodevelopmental disease lissencephaly (*Parrini et al., 2016*; *Reiner et al., 1993*). Structurally, Lis1 is a dimer of two ß-propellers (*Kim et al., 2004*; *Tarricone et al., 2004*), and Lis1 dimerization is required for its function in vivo (*Ahn and Morris, 2001*; *Huang et al., 2012*). Lis1 is also the only known dynein regulator that binds directly to dynein's motor domain (*Huang et al., 2012*; *Toropova et al., 2014*). Structures of yeast dynein–Lis1 complexes, while not of high enough resolution to build molecular models, show that Lis1 binds to dynein at two sites: dynein's AAA+ ring at AAA3/AAA4 ($site_{ring}$) and dynein's stalk ($site_{stalk}$), which emerges from AAA4 and leads to dynein's microtubule binding domain (*Figure 1B*; *DeSantis et al., 2017*; *Htet et al., 2020*; *Huang et al., 2012*; *Toropova et al., 2014*). Lis1 binding to dynein at these two sites is conserved in humans (*Htet et al., 2020*).

How does Lis1 act as a positive regulator of dynein motility at a mechanistic level? Recently, Lis1 was shown to play a role in forming active dynein complexes, most likely by disrupting dynein's auto-inhibited Phi conformation (*Elshenawy et al., 2020*; *Htet et al., 2020*; *Marzo et al., 2020*; *Qiu et al., 2019*). Human dynein–dynactin–activating adaptor complexes formed in the presence of Lis1 are more likely to contain two dynein dimers and move at a faster velocity in vitro (*Elshenawy et al., 2020*; *Htet et al., 2020*), consistent with a previous study showing that Lis1 increases both the frequency and velocity of motile dynein–dynactin–BICD2 complexes (*Baumbach et al., 2017*). Studies in *S. cerevisiae* and the filamentous fungus *Aspergillus nidulans* suggest that this role for Lis1 in forming active complexes is conserved and required for in vivo function (*Marzo et al., 2020*; *Qiu et al., 2019*).

Prior to this series of discoveries, we and others had described other effects that Lis1 has on dynein's mechanochemistry in vitro. It is likely that these in vitro readouts provide mechanistic insights into different intermediates during dynein activation by Lis1. One common theme in these studies was that the presence of Lis1 causes dynein to bind microtubules more tightly under some nucleotide conditions (*DeSantis et al., 2017*; *Htet et al., 2020*; *Huang et al., 2012*; *McKenney et al., 2010*; *Yamada et al., 2008*), which in the context of a moving molecule correlates with decreased velocity (*DeSantis et al., 2017*; *Huang et al., 2012*; *Yamada et al., 2008*) or an increased ability of dynein to remain bound to microtubules under load (*McKenney et al., 2010*). In yeast, the nucleotide state of dynein's AAA3 domain seems to regulate this as Lis1 (called Pac1 in yeast, referred to as 'Lis1' here) induces a weak microtubule binding state in dynein when the nucleotide bound at AAA3 is ATP or an ATP mimic (*DeSantis et al., 2017*). These different modes of dynein regulation by yeast Lis1 correlate with the number of Lis1 ß-propellers observed interacting with dynein by cryo-EM, with a single ß-propeller bound to dynein at $site_{ring}$ in the absence of nucleotide at AAA3, and two ß-propellers bound to dynein at $site_{ring}$ and $site_{stalk}$ when the nucleotide at AAA3 is ATP or an ATP mimic (*DeSantis et al., 2017*). Genetically, mutations that link nucleotide hydrolysis at AAA3 to Lis1 have also been reported (*Qiu et al., 2021*; *Willins et al., 1997*; *Zhuang et al., 2007*). While it remains to be determined how the readouts described above or other in vitro readouts of Lis1 regulation of dynein (*Baumbach et al., 2017*; *Gutierrez et al., 2017*; *Jha et al., 2017*; *Marzo et al., 2020*; *Wang et al., 2013*) relate to

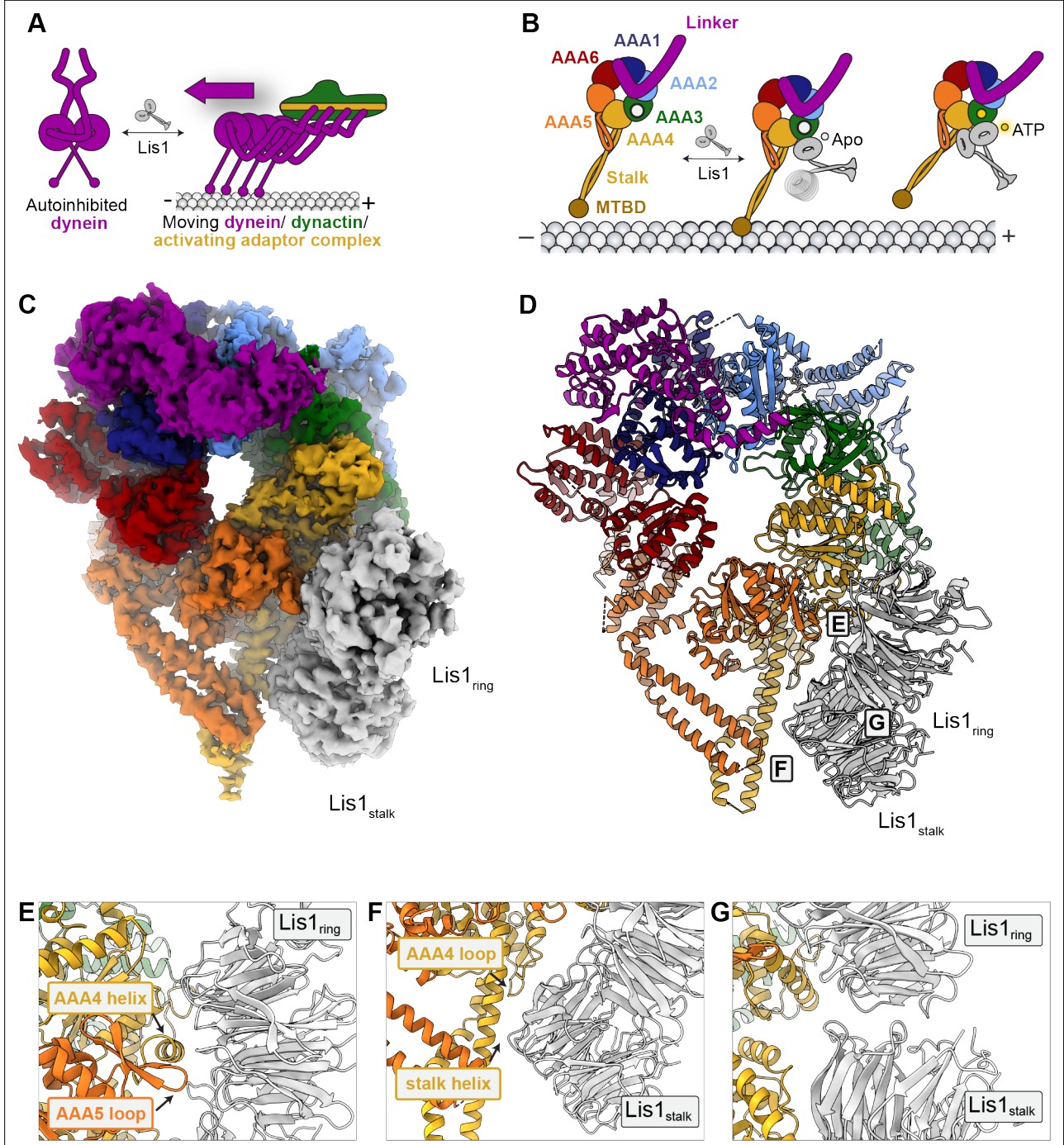

**Figure 1.** A 3.1 Å structure of the yeast dynein$^{E2488Q}$–(Lis1)$_2$ complex. (**A**) Lis1 assists in the transition from autoinhibited dynein to activated dynein complexed with dynactin and an activating adaptor protein. (**B**) Schematic representation of the different modes of binding of Lis1 to dynein as a function of nucleotide state at AAA3. (**C, D**) Cryo-EM map (**C**) and model (**D**) of the AAA3-WalkerB mutant, dynein$^{E2488Q}$, with Lis1 bound at site$_{ring}$ and site$_{stalk}$. (**E–G**) Close-up views of the interfaces between (**E**) dynein$^{E2488Q}$ and Lis1 at site$_{ring}$, (**F**) dynein$^{E2488Q}$ and Lis1 at site$_{stalk}$, and (**G**) of the Lis1$_{ring}$-Lis1$_{stalk}$ interface.

The online version of this article includes the following figure supplement(s) for figure 1:

**Figure supplement 1.** Details of cryo-EM data processing.

**Figure supplement 2.** Examples of cryo-EM map quality for binding sites.

**Figure supplement 3.** 3D variability analysis of dynein–(Lis1)$_2$.

Lis1's in vivo function, they serve as powerful tools to test structure/ function predictions (*DeSantis et al., 2017*; *Toropova et al., 2014*).

Despite this progress, a full understanding of how Lis1 regulates dynein has been hampered by the lack of high-resolution structures of dynein–Lis1 complexes. Here, we report a 3.1 Å structure of yeast dynein bound to two Lis1 ß-propellers. This is the first high-resolution structure of the dynein–Lis1 complex and reveals in molecular detail the contacts between yeast dynein and Lis1 at both $site_{ring}$ and $site_{stalk}$ and the interface between two Lis1 ß-propellers. We use this structure to interrogate the dynein–Lis1 interaction both in vitro and in vivo, showing that these contacts are important for yeast Lis1's role in nuclear positioning in vivo. Finally, we show that mutating $site_{ring}$ and $site_{stalk}$ in human dynein impacts Lis1's ability to promote the formation of activated human dynein complexes, demonstrating that the Lis1 binding sites on dynein that are required for in vivo function in yeast are the same sites that are important for human dynein–dynactin–activating adaptor complex formation. Our results point to a conserved model for dynein regulation by Lis1.

**3.1Å Cryo-EM structure of Dynein–Lis1**

**Video 1.** A 3.1 Å structure of the dynein–Lis1 complex. Overview of the dynein–$(Lis1)_2$ complex. The key interactions between dynein and Lis1 at $site_{ring}$ and $site_{stalk}$, as well as the Lis1–Lis1 interface, are highlighted.
https://elifesciences.org/articles/71229/figures#video1

## Results

### 3.1 Å structure of a yeast dynein–Lis1 complex

Previously, we determined structures of a *S. cerevisiae* dynein motor domain bound to one or two yeast Lis1 ß-propellers by cryo-EM at 7.7 Å and 10.5 Å resolution, respectively (*DeSantis et al., 2017*). Although those structures allowed us to dock homology models of Lis1 into the map and determine their general orientation, their limited resolutions prevented us from building molecular models into the density or rotationally aligning Lis1's ß-propellers. To address these limitations and to understand the molecular features of the interactions between dynein and Lis1, we aimed to improve the overall resolution of this complex. To enrich our sample for complexes containing two Lis1 ß-propellers bound to dynein, we used a construct (*Supplementary file 1*) of the yeast dynein motor domain containing a Walker B mutation in AAA3 (E2488Q; herein referred to as $dynein^{E2488Q}$) that prevents the hydrolysis of ATP as our previous work showed that this state results in Lis1 binding at both $site_{ring}$ and $site_{stalk}$ (*DeSantis et al., 2017*). As with our previous structural studies (*DeSantis et al., 2017*; *Toropova et al., 2014*), we used monomers of the yeast dynein motor domain and yeast Lis1 dimers, referred to as $dynein^{E2488Q}$–$(Lis1)_2$ here. We also added ATP and vanadate to our samples as this is converted to ADP-vanadate, referred to here as ADP-$V_i$, which mimics a post-hydrolysis state for dynein (*Schmidt et al., 2015*). Our initial attempts at obtaining a higher resolution structure were limited by dynein's strong preferred orientation in open-hole cryo-EM grids. To overcome this, we randomly and sparsely biotinylated dynein and applied the sample to streptavidin affinity grids (*Han et al., 2016*; *Han et al., 2012*; *Lahiri et al., 2019*); this tethering results in randomly oriented complexes that are prevented from reaching and interacting with the air–water interface.

This approach allowed us to determine the structure of yeast $dynein^{E2448Q}$ bound to wild-type yeast Lis1 in the presence of ATP vanadate to a nominal resolution of 3.1 Å (*Figure 1C and D*, *Figure 1—figure supplement 1A–C*, *Figure 1—figure supplement 2*, *Video 1*, *Supplementary file 2*). In our structure, $dynein^{E2488Q}$'s AAA ring is 'closed,' which is seen when dynein's microtubule binding domain is in its low-affinity state, the linker is bent (*Bhabha et al., 2014*; *Schmidt et al., 2015*), and when two Lis1 ß-propellers interact with dynein (*DeSantis et al., 2017*). It is likely that the two ß-propellers come from the same homodimer as we did not detect density for additional ß-propellers during data processing. This new high-resolution map allowed us to build an atomic model of the entire

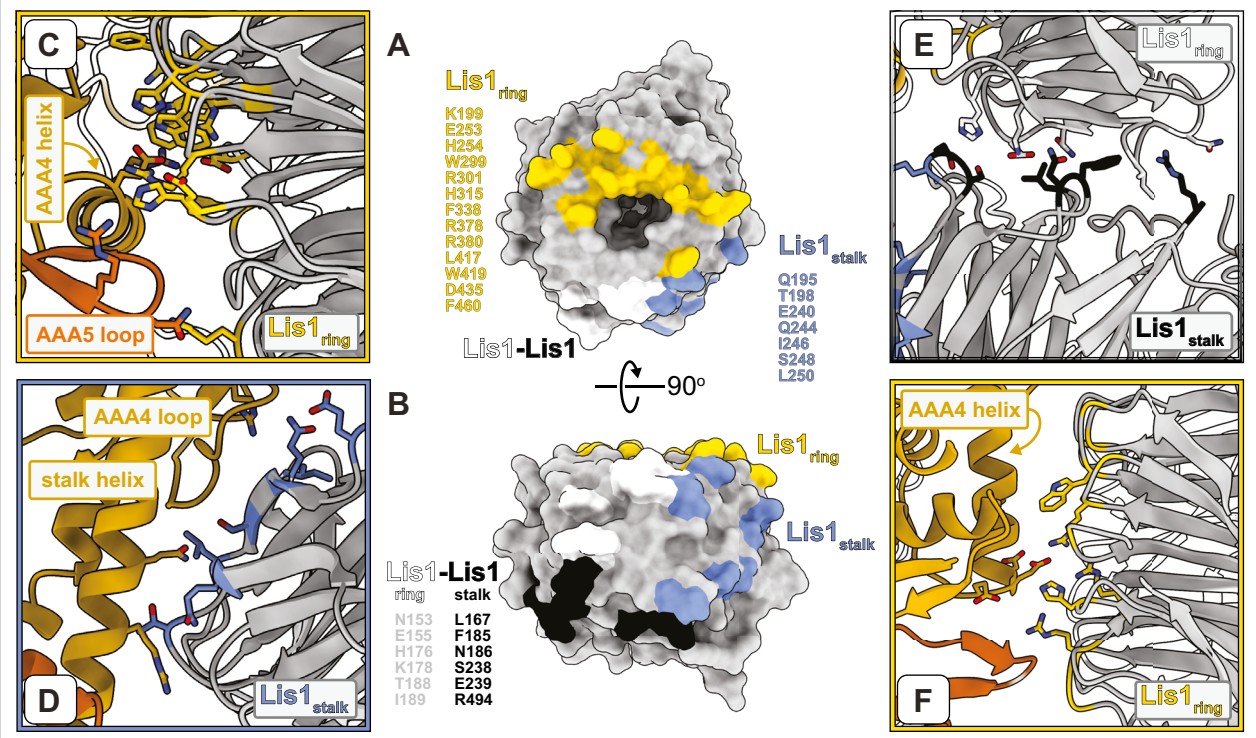

**Figure 2.** Interfaces involved in Lis1–dynein and Lis1–Lis1 interactions. (**A, B**) Two views of the Lis1 structure, represented as a molecular surface. The β-propeller is viewed either from its narrow face (**A**) or the side (**B**). Residues located at the different interfaces formed by Lis1 in the dynein$^{E2488Q}$–(Lis1)$_2$ complex are listed next to and color-coded on the molecular surfaces. We have included all residues within 4 Å of residues in the interacting partner. (**C–F**) Close-ups of the different interfaces highlighted in (**A, B**). Panels (**C**), (**D**), and (**E**) are closer views of panels (**E**), (**F**), and (**G**) in *Figure 1*, in that order. The Lis1 residues listed in (**A, B**) are shown in the panels, along with dynein residues discussed in the text (and shown again in later figures). (**F**) This panel shows the same dynein–Lis1$_{ring}$ interface shown in (**C**) but viewed from 'above' (relative to **C**). In this panel, we show residues we had previously mutated in dynein (*Huang et al., 2012*) and Lis1 (*Toropova et al., 2014*) to disrupt this interaction.

yeast dynein$^{E2488Q}$–(Lis1)$_2$ complex (*Figure 1D*, *Video 1*). Binding of Lis1 at site$_{ring}$ primarily involves the smaller face of the ß-propeller in Lis1 and an alpha helix in AAA4 of dynein$^{E2488Q}$ (*Figures 1E and 2A–C*, *Figure 1—figure supplement 2A*, *Video 2*). Our high-resolution structure also allowed us to unambiguously map an additional interaction between dynein$^{E2488Q}$ and Lis1 at site$_{ring}$ that involves a contact with a loop in AAA5 (*Figures 1E and 2C*, *Figure 1—figure supplement 2B*, *Video 2*). Lis1 binds to dynein at site$_{stalk}$ using its side and makes interactions with coiled-coil 1 (CC1) of dynein's stalk, as well as a previously unknown contact with a loop in AAA4 (residues 2935–2942) (*Figures 1F and 2A, B and D*, *Figure 1—figure supplement 2C*, *Video 2*). As suggested by the continuous density in our previous low-resolution structure (*DeSantis et al., 2017*), the Lis1s bound at site$_{ring}$ and site$_{stalk}$ directly interact with each other (*Figure 1G*, *Figure 1—figure supplement 2D*, *Figure 2A, B and E*, *Video 2*). Our model of the dynein$^{E2488Q}$–(Lis1)$_2$ complex confirmed that residues we had

**Interfaces involved in Dynein-Lis1 and Lis1-Lis1 interactions**

**Video 2.** Interfaces involved in Lis1–dynein and Lis1–Lis1 interactions. Overview of the different interfaces involved in the formation of the dynein–(Lis1)$_2$ complex. This movie accompanies Figure 2 and uses the same color coding. Please refer to Figure 2 and its legend for details.

https://elifesciences.org/articles/71229/figures#video2

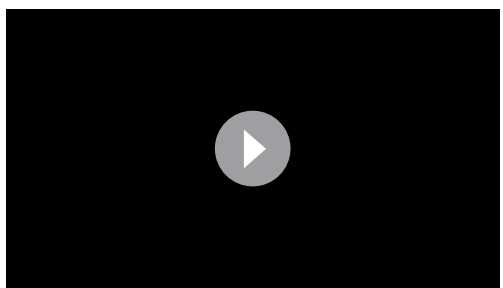

**Video 3.** 3DVA analysis of dynein bound to Lis1. 3DVA analysis reveals conformational changes along the first two variability components. The first variability component describes motion attributed to the linker pulling away from AAA2. The second variability component describes changes in the overall motor domain as it undergoes subtle open and closing motions in the AAA ring.

https://elifesciences.org/articles/71229/figures#video3

previously mutated, either on dynein (*Huang et al., 2012*) or on Lis1 (*Toropova et al., 2014*), to disrupt the site$_{ring}$ interaction based on much lower resolution information, are indeed located at the interface (*Figure 2F*). The ATP binding sites at AAA1, AAA2, AAA3, and AAA4 are all occupied by nucleotides (*Figure 1—figure supplement 2E–H*). While our map shows ADP bound at AAA4 (*Figure 1—figure supplement 2H*), we were unable to resolve whether ATP or ADP-vanadate was bound at AAA1-AAA3 and chose to model ATP into each binding pocket (*Figure 1—figure supplement 2E–H*).

Dynein is a dynamic protein and undergoes multiple conformational changes as part of its mechanochemical cycle. ATP binding and hydrolysis at AAA1 leads to a cascade of conformational changes including closing of the AAA ring, shifting of the buttress and stalk helices, and swinging of the linker domain during its power stroke cycle (*Cianfrocco et al., 2015*). While the AAA ring domain of dynein reached the highest resolution in our structure, the linker and stalk helices are at significantly lower resolutions, most likely due to conformational heterogeneity (*Figure 1—figure supplement 1B*). We used 3D variability analysis (*Punjani and Fleet, 2021*) in cryoSPARC to generate variability components that capture continuous movements within the complex; the first two components describe the major movements. The first variability component shows the linker pulling away from its docked site at AAA2 and starting to straighten towards AAA4 in what could be the beginning of the power stroke (*Figure 1—figure supplement 3A*, *Video 3*). The second variability component shows the AAA ring 'breathing' slightly between its open and closed conformations, with associated movements in the buttress and stalk helices (*Figure 1—figure supplement 3B*, *Video 3*). Importantly, all the interactions involving Lis1—with site$_{ring}$ and site$_{stalk}$ and at the Lis1–Lis1 interface—are maintained in the different conformations.

## Yeast Lis1's interaction with dynein's AAA5 contributes to regulation at site$_{ring}$

Next, we sought to validate the new interactions we identified in our structure of the yeast dynein$^{E2448Q}$–(Lis1)$_2$ complex. To do this, we chose well-established in vitro and in vivo assays with clear readouts for any defects in dynein regulation by Lis1 (*DeSantis et al., 2017*; *Reck-Peterson et al., 2006*). The first site we investigated was the interaction of Lis1 with dynein at site$_{ring}$. The bulk of the yeast dynein$^{E2488Q}$–(Lis1)$_2$ interface at site$_{ring}$ comprised a helix in AAA4 of dynein, mutations in which disrupt yeast Lis1 binding to dynein (*Huang et al., 2012*). Here, we set out to determine if the additional contact we observed at site$_{ring}$ involving AAA5 (*Figure 3A*) also contributes to dynein regulation by Lis1. To disrupt this electrostatic interaction (*Figure 3A*), we either mutated Lis1's Glu253 and His254 to Ala or deleted dynein's Asn3475 and Arg3476, which are both present in a short loop (*Figure 3A*). These mutations were engineered into the endogenous copies of either Lis1 (*PAC1*) or dynein (*DYN1*) in yeast and both were expressed at wild-type levels (*Figure 3—figure supplement 1A–C*). To determine if these mutations had a phenotype in vivo, we performed a nuclear segregation assay. In this assay, deletion of dynein or dynactin subunits or Lis1 causes an increase in binucleate mother cells, which arise from defects in mitotic spindle positioning (*Eshel et al., 1993*; *Kormanec et al., 1991*; *Lee et al., 2003*; *Li et al., 1993*; *Muhua et al., 1994*; *Sheeman et al., 2003*). We found that Lis1$^{E253A, H254A}$ and dynein$^{\Delta N3475, \Delta R3476}$ both exhibit a binucleate phenotype that is equivalent to that seen with deletion of Lis1 or dynein (*Figure 3B–D*), suggesting that the interaction between Lis1 and dynein at AAA5 is required for yeast Lis1's ability to regulate dynein in vivo.

We next characterized the biochemical properties of these mutations designed to disrupt yeast Lis1's interaction with AAA5. We began by measuring the apparent binding affinity between a Lis1

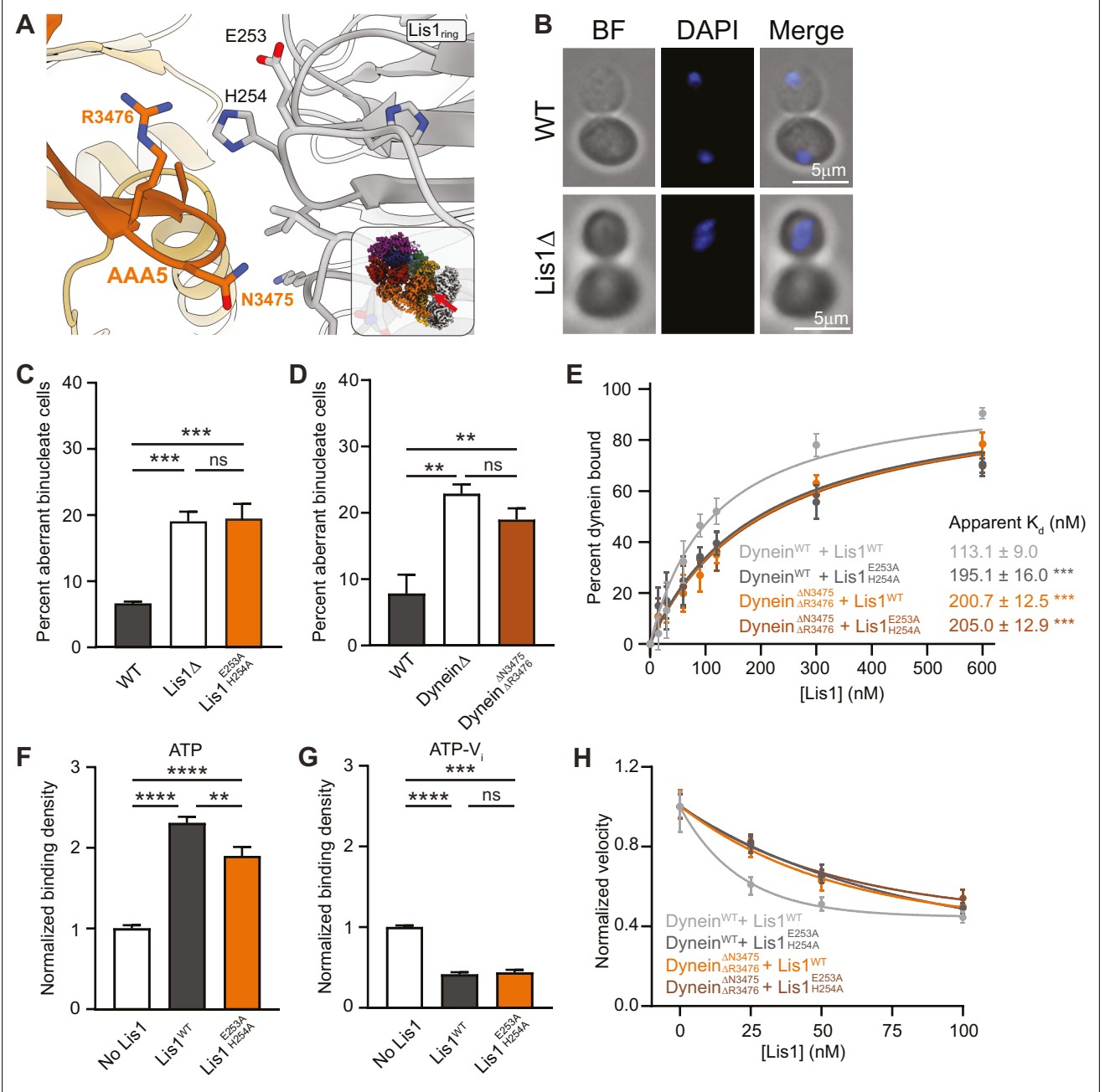

**Figure 3.** Lis1 regulation of dynein at site$_{ring}$. (**A**) Close-up of the interface between Lis1 and dynein AAA5 at site$_{ring}$. The residues mutated in this study on Lis1 (gray type) and dynein (orange type) are indicated. The inset shows the location of the close-up in the full structure. (**B**) Example images of normal and binucleate cells. Scale bar, 3 µm. (**C**) Quantitation (mean ± s.e.m.) of the percentage of cells displaying an aberrant binucleate phenotype for WT (dark gray), Lis1 deletion (white), and Lis1$^{E253A, H254A}$ (orange). n = 6 replicates with at least 200 cells per condition. Statistical analysis was performed with a one-way ANOVA with Tukey's multiple comparison test; ***WT and ΔLis1, p=0.0009; ***WT and Lis1$^{E253A, H254A}$, p=0.0007; ns, p>0.9999. (**D**) Quantitation (mean ± s.e.m.) of the percentage of cells displaying an aberrant binucleate phenotype for WT (dark gray), Lis1 deletion (white), and dynein$^{ΔN3475, R3476}$ (dark orange). n = 3 replicates with at least 200 cells per condition. Statistical analysis was performed with a one-way ANOVA with Tukey's multiple comparison test; **p=0.0053; *p=0.0215; ns, p=0.4343. (**E**) Quantitation of the apparent binding affinity of dynein$^{WT}$ for Lis1$^{WT}$ (dark gray; K$_d$ = 113.1 ± 9.0) and Lis1$^{E253A, H254A}$ (light gray; K$_d$ = 195.1 ± 16.0) and dynein$^{ΔN3475, R3476}$ for Lis1$^{WT}$ (orange; K$_d$ = 200.7 ± 12.5) and Lis1$^{E253A, H254A}$ (brown; K$_d$ = 205.0 ± 12.9). n = 3 replicates per condition. Statistical analysis was performed using an extra sum-of-squares F-test; ***p<0.0001. (**F**) Binding density (mean ± s.e.m.) of dynein on microtubules with ATP in the absence (dark gray) or presence of Lis1$^{WT}$ (white) or Lis1$^{E253A, H254A}$ (orange). Data was normalized to a density of 1.0 in the absence of Lis1. Statistical analysis was performed using an ANOVA; ****p<0.0001; **p=0.0029. n = 12 replicates per condition. (**G**) Binding density (mean ± s.e.m.) of wild-type dynein on microtubules with ATP-V$_i$ in the absence (dark gray) or presence of Lis1$^{WT}$ (white) or Lis1$^{E253A, H254A}$ (orange). Data was normalized to a density of 1.0 in the absence of Lis1. Statistical analysis was performed using an ANOVA;

*Figure 3 continued on next page*

*Figure 3 continued*

****p<0.0001; ***p=0.0001; ns = 0.9997. n = 12 replicates per condition. (**H**) Single-molecule velocity of dynein with increasing concentrations of Lis1. Dynein$^{WT}$ with Lis1$^{WT}$ (dark gray); dynein$^{WT}$ with Lis1$^{E253A, H254A}$ (light gray); dynein$^{\Delta N3475, R3476}$ with Lis1$^{WT}$ (orange); dynein$^{\Delta N3475, R3476}$ and Lis1$^{E253A, H254A}$ (brown). The median and interquartile range are shown. Data was normalized to a velocity of 1.0 in the absence of Lis1. At least 400 single-molecule events were measured per condition.

The online version of this article includes the following source data and figure supplement(s) for figure 3:

**Figure supplement 1.** Western blot analysis of Lis1 and dynein mutant expression.

**Figure supplement 1—source data 1.** Western blot analysis of Lis1 and dynein mutant expression.

dimer and a monomeric dynein motor domain. For these experiments, purified SNAP-tagged yeast Lis1 was covalently coupled to magnetic beads, mixed with purified yeast dynein motor domains in the presence of ADP, and the percentage of dynein bound to the beads was quantified. We found that mutations at the Lis1–AAA5 interface, either on Lis1 (Lis1$^{E253A, H254A}$) or on dynein (dynein$^{\Delta N3475, \Delta R3476}$), led to an ~1.75-fold decrease in the affinity of Lis1 for dynein (*Figure 3E*). Next, we looked at Lis1's effect on dynein's interaction with microtubules. Previously, we showed that tight microtubule binding by dynein in the presence of ATP is mediated by site$_{ring}$, whereas Lis1-induced weak microtubule binding in the presence of ATP-V$_i$ requires both site$_{ring}$ and site$_{stalk}$ (*DeSantis et al., 2017*). For these experiments, we used purified yeast Lis1 dimers and truncated and GST-dimerized yeast dynein motors (*Reck-Peterson et al., 2006*). These GST-dynein dimers have been well characterized and behave similarly to full-length yeast dynein in vitro (*Gennerich et al., 2007*; *Reck-Peterson et al., 2006*). As expected for a mutation that would disrupt Lis1's interaction at site$_{ring}$, we found that Lis1$^{E253A, H254A}$ decreased Lis1's ability to enhance the binding of dynein to microtubules in the presence of ATP (*Figure 3F*). In contrast, Lis1$^{E253A, H254A}$ did not affect Lis1's ability to lower dynein's affinity for microtubules in the presence of ATP-V$_i$ (*Figure 3G*).

Finally, we measured the effect of disrupting the Lis1–AAA5 interaction on the velocity of dynein using a single-molecule motility assay. Unlike mammalian dynein, *S. cerevisiae* dynein moves processively in vitro in the absence of any other dynein subunits or cofactors (*Reck-Peterson et al., 2006*). We previously showed that addition of Lis1 to in vitro motility assays slows yeast dynein in a dose-dependent manner (*Huang et al., 2012*). Here, we found that in the absence of Lis1, dynein$^{\Delta N3475, \Delta R3476}$ had a similar velocity to wild-type dynein, indicating that the mutation does not affect the motile properties of the motor (*Figure 3H*). In contrast, Lis1 was less effective at slowing the velocity of dynein$^{\Delta N3475, \Delta R3476}$ compared to wild-type dynein. We observed a similar effect with Lis1$^{E253A, H254A}$, which targets the same site$_{ring}$ interface, but from the Lis1 side (*Figure 3H*). Thus, the interaction of yeast Lis1 with dynein at AAA5 contributes to Lis1's regulation of dynein at site$_{ring}$.

## Identification of yeast Lis1 mutations that specifically disrupt regulation at site$_{stalk}$

After interrogating the interaction between yeast Lis1 and dynein at site$_{ring}$, we turned to site$_{stalk}$. Previously, we identified amino acids in yeast dynein's stalk (E3012, Q3014, and N3018) that were required for the Lis1-mediated decrease in dynein's affinity for microtubules in the presence of ATP-V$_i$ (*DeSantis et al., 2017*). However, the resolution of our previous structure was not high enough to accurately dock in a homology model of the yeast Lis1 ß-propeller at this site. Our new structure revealed this interaction in molecular detail, showing a new contact between yeast Lis1 and dynein's site$_{stalk}$ involving a short loop in AAA4 of dynein (*Figure 4A*). To determine if this contact plays a role in yeast dynein regulation by Lis1, we mutated Ser248 in the endogenous copy of Lis1 to Gln to introduce a steric clash with the loop in dynein's AAA4 (*Figure 4A*). In a nuclear segregation assay, yeast cells expressing Lis1$^{S248Q}$ at wild-type levels (*Figure 3—figure supplement 1*) showed a defect in nuclear segregation (*Figure 4B*). Thus, this new contact at site$_{stalk}$ between AAA4 of dynein and Lis1 is also required for Lis1's ability to regulate dynein in vivo.

We next probed the role of this new contact in vitro. Since Ser248 is only involved in the interaction between Lis1 and site$_{stalk}$, Lis1$^{S248Q}$ should maintain its ability to bind to and regulate site$^{ring}$. In agreement with this, we found that Lis1$^{S248Q}$ showed a moderate decrease in binding affinity for dynein monomers (*Figure 4C*). Similarly, Lis1$^{S248Q}$ remained capable of inducing tight microtubule binding in the presence of ATP (regulation at site$_{ring}$) (*Figure 4D*), but failed to decrease microtubule binding

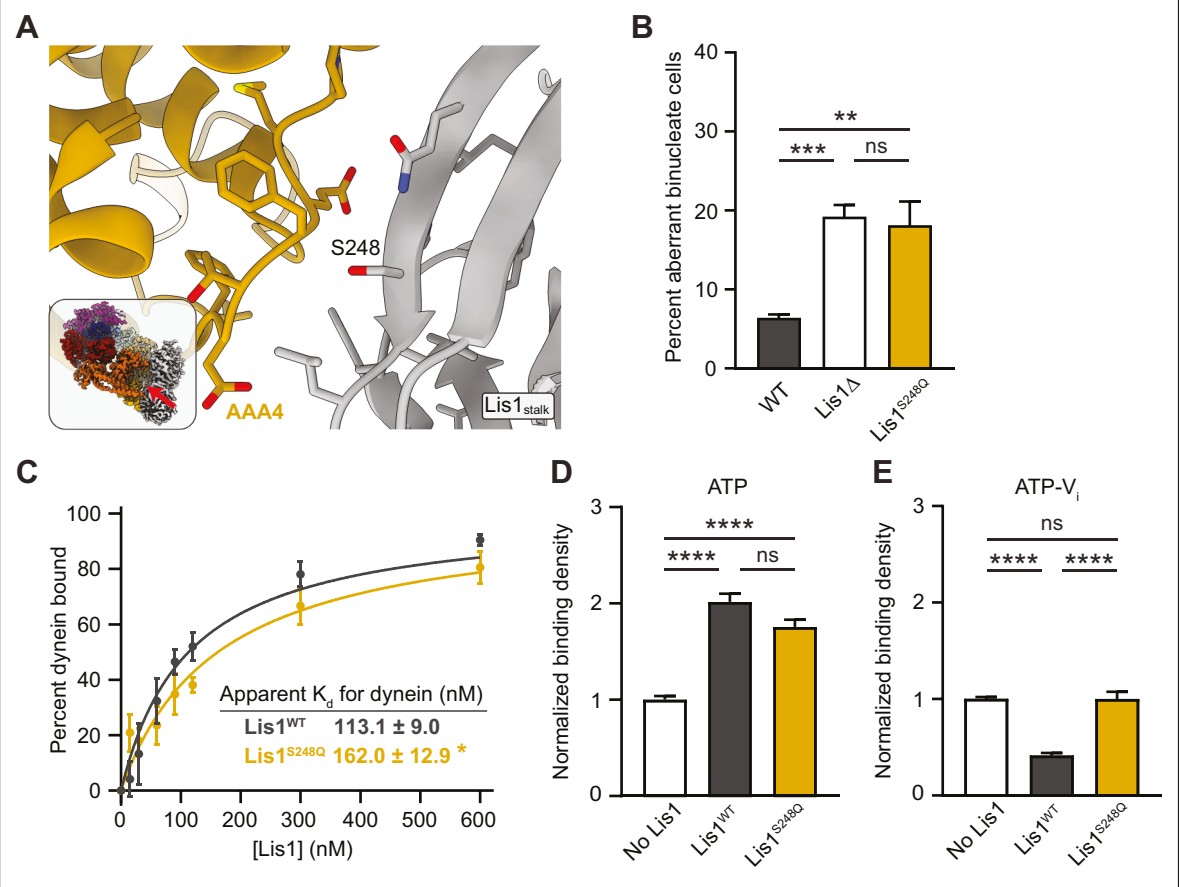

**Figure 4.** Lis1 regulation of dynein at site$_{stalk}$. (**A**) Close-up view of the Lis1–dynein interface at site$_{stalk}$. The residues at the interface are shown, and the S248 mutated in this study is labeled. The inset shows the location of the close-up in the full structure. (**B**) Quantitation of the percentage of cells (mean ± s.e.m.) displaying an aberrant binucleate phenotype for WT (dark gray), Lis1 deletion (white), and Lis1$^{S248Q}$ (yellow). n = 6 replicates of at least 200 cells each per condition. Statistical analysis was performed with a one-way ANOVA with Tukey's multiple comparison test; ***p=0.0009; **p=0.0028; ns, p=0.9908. WT and ΔLis1 data repeated from *Figure 2*. (**C**) Quantitation of the apparent binding affinity of dynein for Lis1$^{WT}$ (dark gray; K$_d$ = 113.1 ± 9.0) and Lis1$^{S248Q}$ (yellow; K$_d$ = 162.0 ± 12.9). n = 3 replicates per condition. Statistical analysis was performed using an extra sum-of-squares *F*-test; p=0.0022. (**D**) Binding density (mean ± s.e.m.) of wild-type dynein on microtubules with ATP in the absence (white) or presence of Lis1$^{WT}$ (dark gray) or Lis1$^{S248Q}$ (yellow). Data was normalized to a density of 1.0 in the absence of Lis1. Statistical analysis was performed using an ANOVA; ****p<0.0001; ns, p=0.0614. n = 12 replicates per condition. (**E**) Binding density (mean ± s.e.m.) of wild-type dynein on microtubules with ATP-V$_i$ in the absence (white) or presence of Lis1$^{WT}$ (dark gray) or Lis1$^{S248Q}$ (yellow). Data was normalized to a density of 1.0 in the absence of Lis1. Statistical analysis was performed using an ANOVA; ****p<0.0001; ns, p>0.9999. n = 12 replicates per condition.

in the presence of ATP-V$_i$ (loss of regulation at site$_{stalk}$) (*Figure 4E*). This makes Lis1$^{S248Q}$ the first Lis1 mutant capable of specifically targeting regulation at site$_{stalk}$.

## The interaction between the two yeast Lis1 ß-propellers is required for regulation at site$_{stalk}$

The resolution of our new structure allowed us to build models for yeast Lis1, revealing the interface between the two Lis1 ß-propellers when they are bound to dynein at both site$_{ring}$ and site$_{stalk}$ (*Figure 5A*). The Lis1–Lis1 interface is moderately hydrophobic with a few electrostatic interactions. To test if this interface is required for dynein regulation in yeast, we aimed to disrupt it with the following mutations: F185D and I189D to introduce charge clashes, and R494A to remove a hydrogen bond. In a nuclear segregation assay, Lis1$^{F185D, I189D, R494A}$, which is expressed at wild-type levels (*Figure 3—figure supplement 1*), resulted in a binucleate yeast cell phenotype that was comparable to that seen with deletion of Lis1 (*Figure 5B*). Thus, the interaction between Lis1 β-propellers is required for Lis1's ability to regulate dynein in vivo.

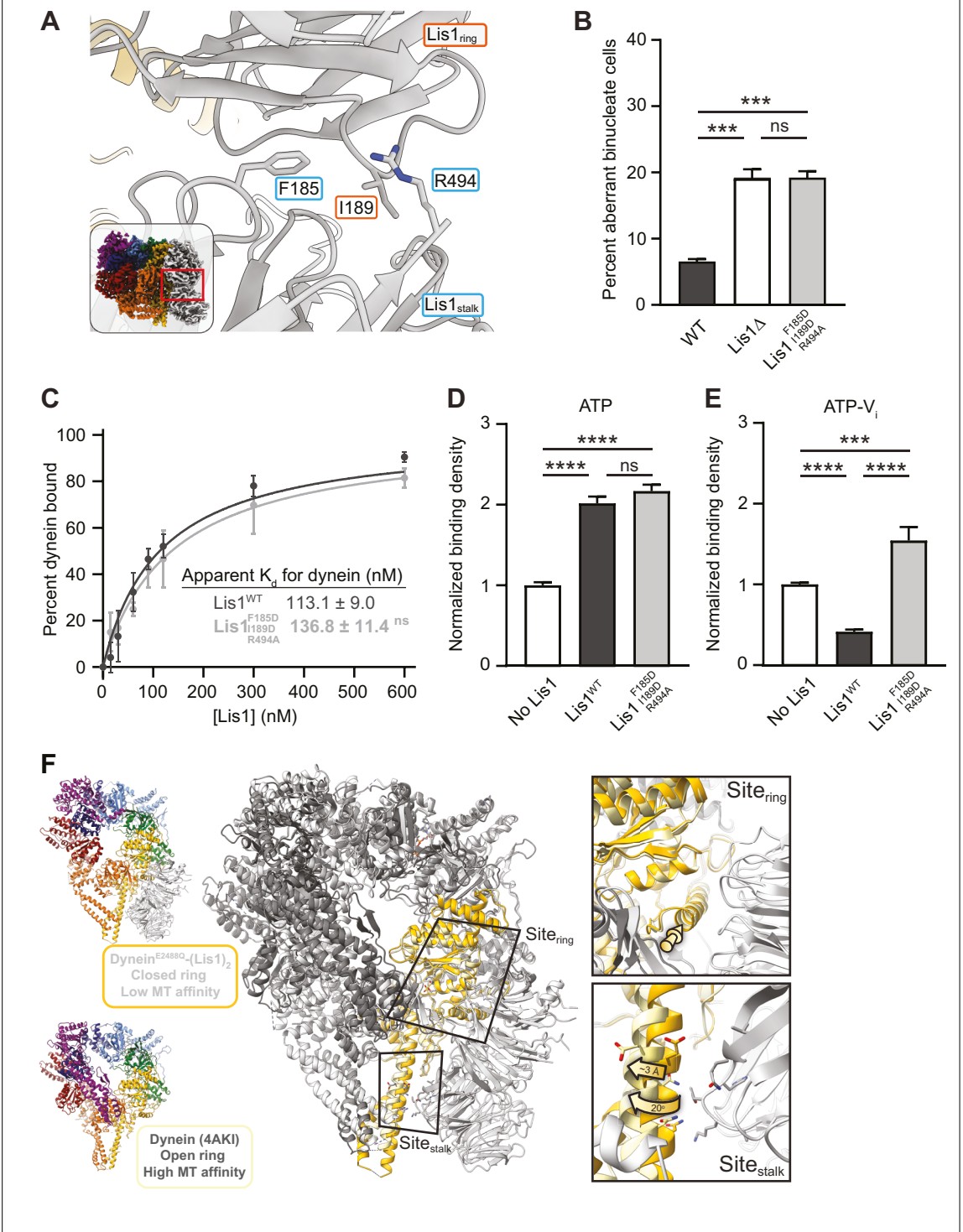

**Figure 5.** Role of the Lis1$_{ring}$–Lis1$_{stalk}$ interaction in Lis1's regulation of dynein. (**A**) Close-up of the interface between the two Lis1 ß-propellers. The residues mutated in this study are indicated. The inset shows the location of the close-up in the full structure. (**B**) Quantitation (mean ± s.e.m.) of the percentage of cells displaying an aberrant binucleate phenotype for WT (dark gray), ΔLis1 (white), and Lis1$^{F185D, I189D, R494A}$ (light gray). n = 6 replicates of at least 200 cells each per condition. Statistical analysis was performed with a one-way ANOVA with Tukey's multiple comparison test; *** WT and ΔLis1, p=0.0009; ***WT and Lis1 $^{F185D, I189D, R494A}$, p=0.0008; ns, p>0.9999. WT and ΔLis1 data repeated from *Figure 2*. (**C**) Quantitation of the binding affinity of dynein for Lis1$^{WT}$ (dark gray; K$_d$ = 113.1 ± 9.0) and Lis1$^{F185D, I189D, R494A}$ (light gray; K$_d$ = 136.8 ± 11.4). n = 3 replicates per condition. Statistical analysis was performed using an extra sum-of-squares *F*-test; p=0.0974. Binding affinity of dynein for Lis1$^{WT}$ repeated from *Figure 2*. (**D**) Binding density (mean ± s.e.m.) of wild-type dynein on microtubules with ATP in the absence (white) or presence of Lis1$^{WT}$ (dark gray) or Lis1$^{F185D, I189D, R494A}$ (light gray).

*Figure 5 continued on next page*

Figure 5 continued

Data was normalized to a density of 1.0 in the absence of Lis1. Statistical analysis was performed using an ANOVA; ****p<0.0001; ns, p=0.4445. n = 12 replicates per condition. (E) Binding density (mean ± s.e.m.) of wild-type dynein on microtubules with ATP-$V_i$ in the absence (dark gray) or presence of Lis1$^{WT}$ (white) or Lis1$^{F185D, I189D, R494A}$ (light gray). Data was normalized to a density of 1.0 in the absence of Lis1. Statistical analysis was performed using an ANOVA; ****p<0.0001; ***p=0.0002. n = 12 replicates per condition. (F) Comparison between the structures of the dynein–(Lis1)$_2$ complex (this work; dynein colored light gray and dark yellow), where dynein's ring is in its closed conformation (low affinity for the microtubule), and of dynein with its ring in the open conformation (high affinity for the microtubule; PDB: 4AKI; dynein colored dark gray and light yellow; *Schmidt et al., 2012*; left). The two structures were aligned using the globular portion of AAA4, where site$_{ring}$ is located, and the two Lis1 ß-propellers from the dynein–(Lis1)$_2$ structure are shown. Close-ups of site$_{ring}$ and site$_{stalk}$ (right) show that while the alpha helix to which Lis1 binds at site$_{ring}$ is in a very similar position in both structures (top, yellow arrow), site$_{stalk}$ has shifted by ~3 Å and rotated by 20° away from where Lis1$_{Stalk}$ would be located (bottom).

In vitro, Lis1$^{F185D, I189D, R494A}$ has an affinity for dynein comparable to that of wild-type Lis1 (*Figure 5C*). In microtubule binding assays, this mutant showed no defects in its ability to stimulate tight microtubule binding by dynein in the presence of ATP (regulation at site$_{ring}$) (*Figure 5D*). In contrast, not only did Lis1$^{F185D, I189D, R494A}$ lose its ability to decrease dynein's affinity for microtubules in the presence of ATP-$V_i$ (loss of regulation at site$_{stalk}$), it increased it (*Figure 5E*). This behavior was most similar to what we had previously observed when we mutated dynein's site$_{stalk}$ (E3012A, Q3014A, N3018A), a mutation that also led to a Lis1-dependent increase in microtubule binding in the presence of ATP-$V_i$ (*DeSantis et al., 2017*). Thus, our data show that regulation of dynein at site$_{stalk}$ requires both binding at site$_{stalk}$, which would not be disrupted in Lis1$^{F185D, I189D, R494A}$, and the interaction between Lis1's ß-propellers, which would.

Given the importance of the Lis1–Lis1 interaction for Lis1's regulation of dynein at site$_{stalk}$, we wondered what prevents Lis1 from binding there when dynein's ring is in its open conformation (high affinity for the microtubule) as our data showed that this interaction plays no role in Lis1's regulation at site$_{ring}$. A comparison of our structure of dynein$^{E2488Q}$ –(Lis1)$_2$ (with dynein's ring in its closed conformation) with that of yeast dynein in an open-ring conformation (PDB: 4AKI; *Schmidt et al., 2012*) showed that while site$_{ring}$ is very similar in both structures, site$_{stalk}$ has shifted ~3 Å and rotated 20° away from where Lis1$_{stalk}$ would be located (*Figure 5F*). This suggests that the interaction between the two Lis1 ß-propellers prevents Lis1 from binding to dynein at site$_{stalk}$ when dynein is in its open-ring state (high affinity for the microtubule), as we had hypothesized previously (*DeSantis et al., 2017*).

## Yeast dynein–Lis1 and Lis1–Lis1 contacts are required for dynein to reach the cortex

Next, we wanted to determine how Lis1 binding to dynein at site$_{ring}$ and site$_{stalk}$, as well as the Lis1–Lis1 interface, contributed to dynein's localization in yeast cells (*Figure 6A and B*). Yeast dynein is active at the cell cortex (*Figure 6A*, right panel), where it pulls on spindle pole body (SPB)-anchored microtubules to position the mitotic spindle (*Adames and Cooper, 2000*). Dynein reaches the cortex by localizing to microtubule plus ends either via kinesin-dependent transport or recruitment from the cytosol (*Carvalho et al., 2004*; *Caudron et al., 2008*; *Markus et al., 2009*). In yeast, Lis1 is required for dynein's localization to SPBs, microtubule plus ends, and the cell cortex (*Lee et al., 2003*; *Li et al., 2005*; *Markus et al., 2009*; *Markus and Lee, 2011*; *Sheeman et al., 2003*). To interrogate dynein's localization in each of the Lis1 mutant backgrounds, the endogenous copy of dynein was labeled at its carboxy terminus with 3xGFP in cells with fluorescently labeled microtubules (*CFP-TUB1*) and SPBs (*SPC110-tdTomato*) (*Markus et al., 2009*; *Figure 6C*, *Figure 6—figure supplement 1*). We then determined the number of dynein foci per cell that were found at SPBs (*Figure 6D*), microtubule plus ends (*Figure 6E*), or the cell cortex (*Figure 6F*). As observed previously, deletion of Lis1 causes a significant decrease in dynein foci at all three sites (*Figure 6D–F*). Mutation of site$_{ring}$ (Lis1$^{E253A, H254A}$) also showed a significant decrease in dynein foci at all three sites and was not significantly different than the Lis1 deletion strain (*Figure 6D–F*). Mutation of site$_{stalk}$ (Lis1$^{S248Q}$) or the Lis1–Lis1 interface (Lis1$^{F185D, I189D, R494A}$) showed intermediate phenotypes for dynein localization to SPBs or the cortex (*Figure 6D and F*). In contrast, the site$_{stalk}$ mutant is the only Lis1 mutant with no defect in dynein's localization to microtubule plus ends (*Figure 6E*), making Lis1$^{S248Q}$ a separation-of-function mutant that could be a useful tool to further understand the mechanism of dynein regulation by Lis1. Overall, our analysis of these new Lis1 mutants supports a role for site$_{ring}$, site$_{stalk}$ and the Lis1–Lis1 interface in localizing dynein to the cell cortex, where active dynein is complexed with dynactin and Num1.

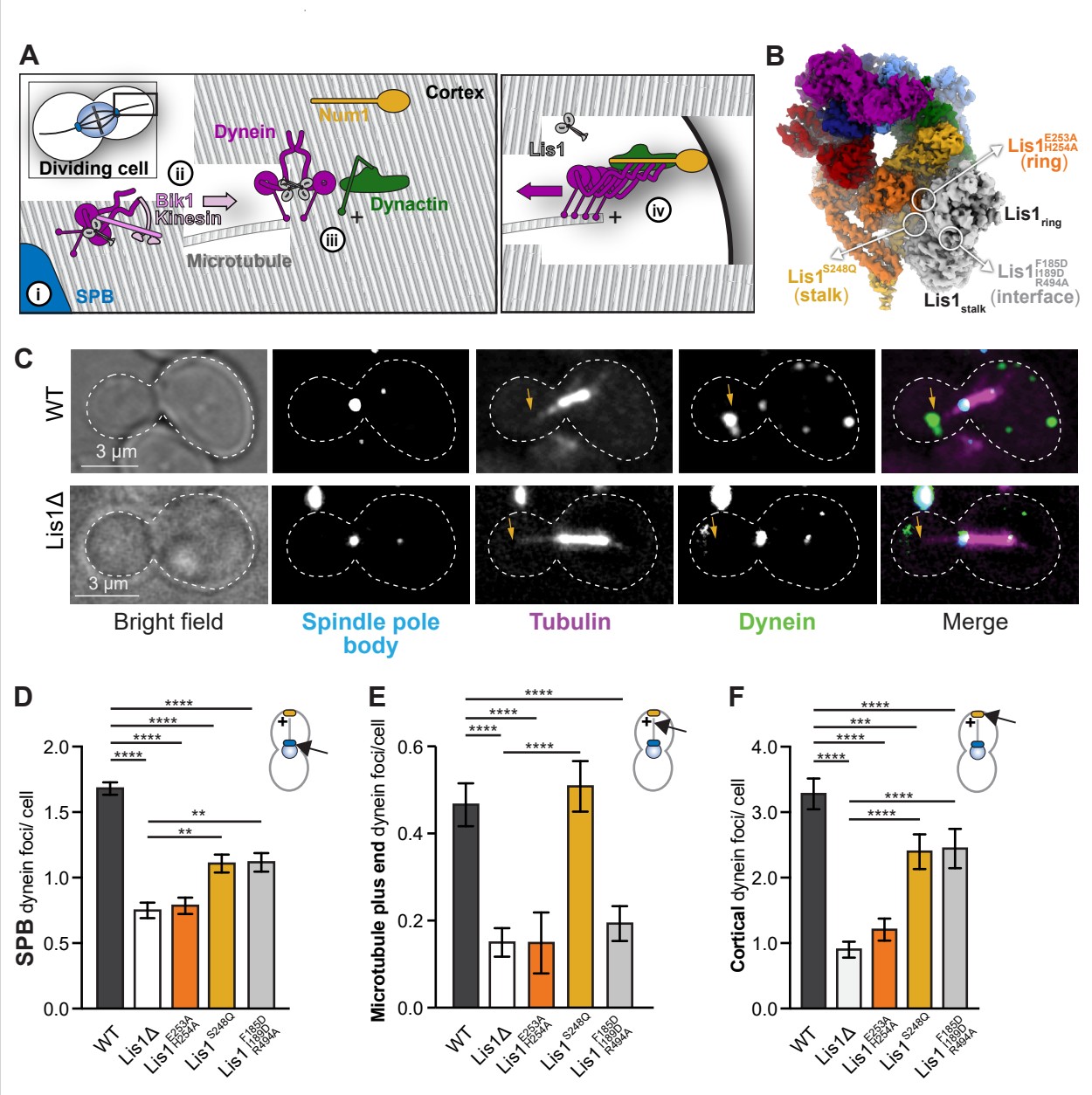

**Figure 6.** Lis1 binding sites on dynein have different roles in the localization of active dynein complexes. (**A**) Schematic showing the sites of dynein localization in dividing yeast cells. At steady state, dynein is found at the spindle pole body (SPB [i]), the microtubule plus end (ii), and the cell cortex (iii). Cortex-associated dynein, which is associated with dynactin and the putative activating adaptors Num1, is active and pulls on SPB-attached microtubules to position the mitotic spindle (iv). (**B**) Our structure showing the location of the mutations that disrupt Lis1 binding at site$_{ring}$, site$_{stalk}$, and the Lis1:Lis1 interface. (**C**) Example images of dynein localization in dividing yeast cells. Orange arrowheads point to microtubule plus ends. (**D–F**) Quantification of dynein localization. Bar graphs show the average number of dynein foci per cell localized to the SPB (**D**), microtubule plus end (**E**), and cortex (**F**) in wild type, Lis1Δ, Lis1$^{E253A, H254A}$, Lis1$^{S248Q}$, and Lis1$^{F185D, I189D, R494A}$ yeast strains. Statistical analysis was performed using a Kruskal–Wallis test; ****$p<0.0001$; ***$p=0.0001$; ns, $p>0.9999$. n = 120 cells per condition.

The online version of this article includes the following figure supplement(s) for figure 6:

**Figure supplement 1.** Yeast dynein–Lis1 and Lis1–Lis1 contacts are required for dynein to reach the cortex.

## Yeast Lis1 regulates dynein beyond the relief of its autoinhibited state

Recent experiments revealed that dynein mutations that disrupt its ability to form the autoinhibited Phi conformation genetically rescue phenotypes seen when Lis1 is mutated or deleted (*Marzo et al., 2020*; *Qiu et al., 2019*), suggesting that a crucial in vivo role for Lis1 is to relieve dynein

autoinhibition. To understand what other roles Lis1 may have in vivo, we made a Phi-disrupting dynein mutant (dynein$^{D2868K}$) (*Marzo et al., 2020*) in the endogenous copy of yeast dynein and compared its phenotype to wild-type cells, cells bearing a Lis1$^{E253A, H254A}$ mutation (which disrupts Lis1 binding at site$_{ring}$), and cells with both the dynein$^{D2868K}$ and Lis1$^{E253A, H254A}$ mutations. We chose to perform this

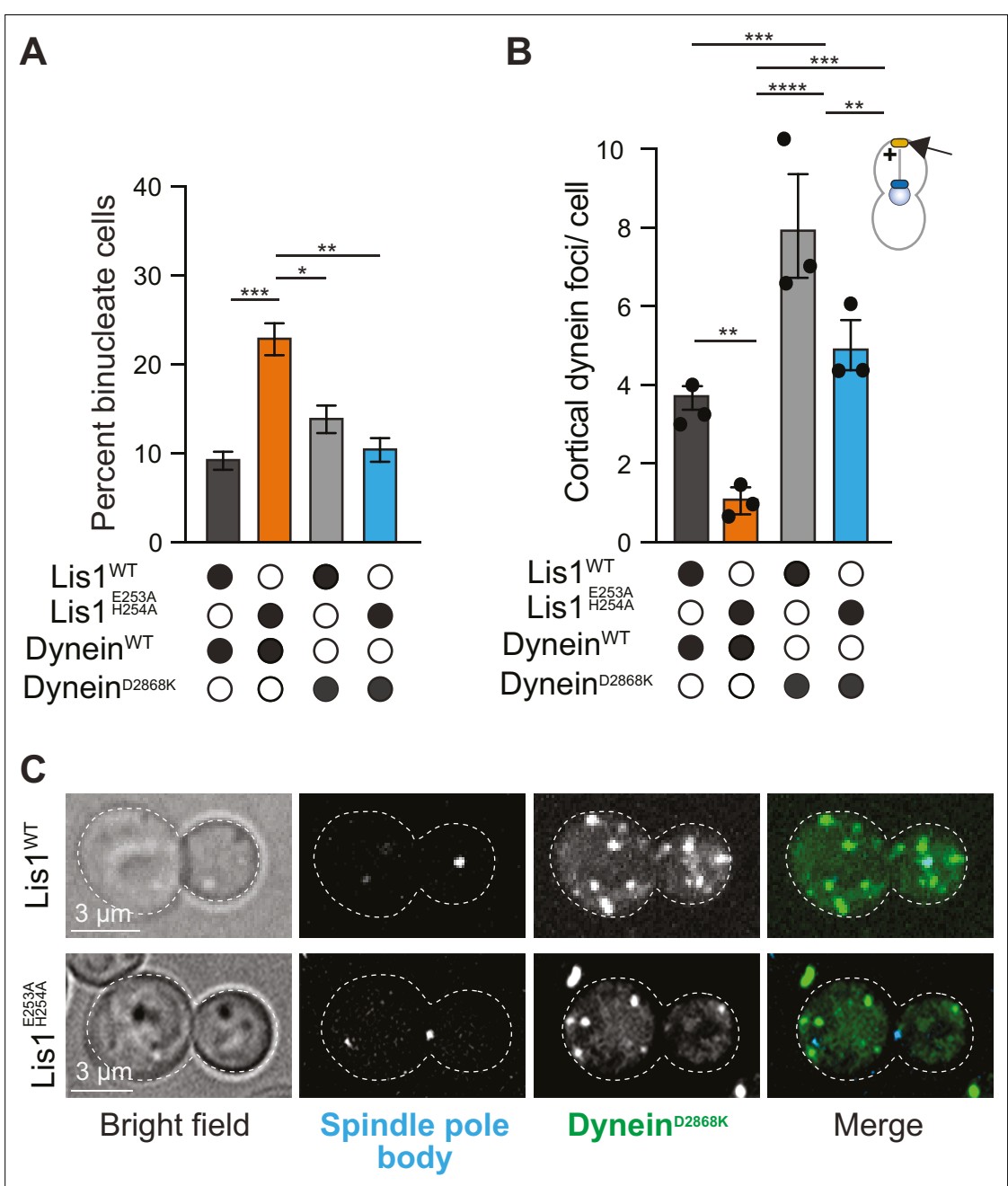

**Figure 7.** Yeast Lis1 regulates dynein beyond the relief of its autoinhibited state. (**A**) Quantitation of the percentage of cells (mean ± s.e.m.) displaying an aberrant binucleate phenotype for WT dynein and Lis1 (dark gray), WT dynein and Lis1$^{E253A, H254A}$ (orange), dynein mutated to relieve Phi autoinhibition (dynein$^{D2868K}$) and Lis1 (gray), and dynein$^{D2868K}$, Lis1$^{E253A, H254A}$ (blue). n = 4 replicates of at least 200 cells per condition. Statistical analysis was performed with a one-way ANOVA with Tukey's multiple comparison test; ***p=0.001; **p=0.0025; *p=0.036; all comparisons that are not labeled were not significant. (**B**) Quantification of dynein localization to the cortex. Bar graphs show the average number of dynein foci per cell localized to the cortex in WT dynein and Lis1 (dark gray), WT dynein and Lis1$^{E253A, H254A}$ (orange), Dynein$^{D2868K}$ and Lis1 (gray), and Dynein$^{D2868K}$ and Lis1$^{E253A, H254A}$ (blue). Statistical analysis was performed with a one-way ANOVA with Tukey's multiple comparison test; ****p<0.0001; ***p=0.0002, and p=0.0003; **p=0.0028 and 0.0025; all comparisons that are not labeled were not significant. n = 120 cells per condition, three replicates. (**C**) Example images of dynein$^{D2868K}$-3xGFP localization in Lis$^{WT}$ and Lis1$^{E E253A, H254A}$ dividing yeast cells.

analysis with the site$_{ring}$ Lis1 mutation because this site regulates tight microtubule binding in vitro. First, we examined the in vivo phenotype of these strains by performing nuclear segregation assays. As reported above, Lis1$^{E253A, H254A}$ increases the percentage of binucleate cells (*Figures 3C and 7A*). However, introducing the dynein Phi-disrupting mutation (dynein$^{D2868K}$) into this background rescues the phenotype (*Figure 7A*). Does Lis1 have any additional roles in regulating yeast dynein in vivo beyond relieving its autoinhibited state? To address this, we examined dynein's localization in live yeast cells bearing the dynein$^{D2868K}$ or Lis1$^{E253A, H254A}$ mutations, or both, and quantified the number of dynein cortical foci per cell (*Figure 7B and C*). As reported previously (*Marzo et al., 2020*), we found that the Phi-disrupting dynein mutation, dynein$^{D2868K}$, results in a significant increase in dynein foci at the cell cortex. In contrast, when dynein$^{D2868K}$ was combined with Lis1$^{E253A, H254A}$ we found that the number of dynein cortical foci per cell was decreased relative to dynein$^{D2868K}$ alone (*Figure 7B and C*). This suggests that binding of Lis1 to dynein at site$_{ring}$ has an additional role in regulating dynein that is downstream of relieving dynein autoinhibition. We note that in published work using more sensitive in vivo assays dynein mutants that relieve autoinhibition cannot fully suppress loss of Lis1 in *A. nidulans* or *S. cerevisiae* (*Marzo et al., 2020*; *Qiu et al., 2019*), further supporting the idea that Lis1 has additional roles beyond relieving dynein's autoinhibition.

## site$_{ring}$ and site$_{stalk}$ are important for Lis1's role in assembling active human dynein complexes

Next, we wondered whether site$_{ring}$ and site$_{stalk}$ were also important for dynein to form complexes with dynactin and an activating adaptor. To do this, we turned to an assay using human proteins, where Lis1 enhances the formation of dynein–dynactin–activating adaptor complexes that contain two dynein dimers, which move faster than complexes containing a single dynein dimer (*Elshenawy et al., 2020*; *Grotjahn et al., 2018*; *Htet et al., 2020*; *Urnavicius et al., 2018*; *Figure 1A*). To determine if human Lis1 binding to human dynein at site$_{ring}$ and site$_{stalk}$ was important for forming these fast-moving complexes, we mutated human dynein at both site$_{ring}$ (K2898A, E2902G, E2903S, and E2904G) and site$_{stalk}$ (E3196A, Q3198A, and N3202A) (*Figure 8A*); we will refer to this mutant as dynein$^{mut}$. We purified wild-type and mutant human dynein from baculovirus-infected insect cells expressing all of the dynein chains (*Schlager et al., 2014*) and measured the affinity of dynein$^{WT}$ and dynein$^{mut}$ for human Lis1. For these experiments, purified human Halo-tagged Lis1 was covalently coupled to magnetic beads, mixed with purified human dynein, and the percentage of dynein bound to the beads was quantified. Lis1 bound to dynein$^{mut}$ with an apparent $K_d$ that was nearly threefold weaker than that for dynein$^{WT}$ (*Figure 8B*). The fact that these mutations do not completely abolish the dynein–Lis1 interaction raises that possibility that there are additional sites of dynein–Lis1 interaction and/or that there are other contacts between dynein and Lis1 at site$_{ring}$ and site$_{stalk}$ in the human system that are not conserved in yeast.

We next examined the single-molecule motility properties of dynein$^{WT}$ and dynein$^{mut}$, complexed with human dynactin and the activating adaptor Hook3. Complexes containing dynein$^{mut}$ moved at the same velocity and showed the same percentage of processive runs when compared to dynein$^{WT}$ in the absence of Lis1 (*Figure 8C and D*). As was shown previously (*Elshenawy et al., 2020*; *Htet et al., 2020*), the velocity of dynein$^{WT}$ was significantly increased by Lis1 (*Figure 8C*), as were the number of processive runs (*Figure 8D*). In contrast, dynein$^{mut}$'s velocity was only modestly increased by Lis1 (*Figure 8C*) and the percentage of processive runs did not significantly increase (*Figure 8D*). We observed a similar trend with dynein–dynactin complexes activated by BicD2 (*Figure 8—figure supplement 1A*). These results mirror our previous findings using a Lis1 mutant containing five point mutations (Lis1$^{5A}$: R212A, R238A, R316A, W340A, K360A) (*Htet et al., 2020*). Here, we repeated this experiment using the same proteins we used for the dynein$^{mut}$ experiments and find that Lis1$^{5A}$ and dynein$^{mut}$ have similar defects in Lis1 regulation of dynein (*Figure 8—figure supplement 1B and C*). These data indicate that site$_{ring}$ and site$_{stalk}$ are important for Lis1's role in forming the activated human dynein–dynactin–activating adaptor complex.

## Discussion

Previous structures of yeast dynein–Lis1 complexes (*DeSantis et al., 2017*; *Toropova et al., 2014*) were not of high enough resolution to reliably fit homology models based on the human Lis1 structure

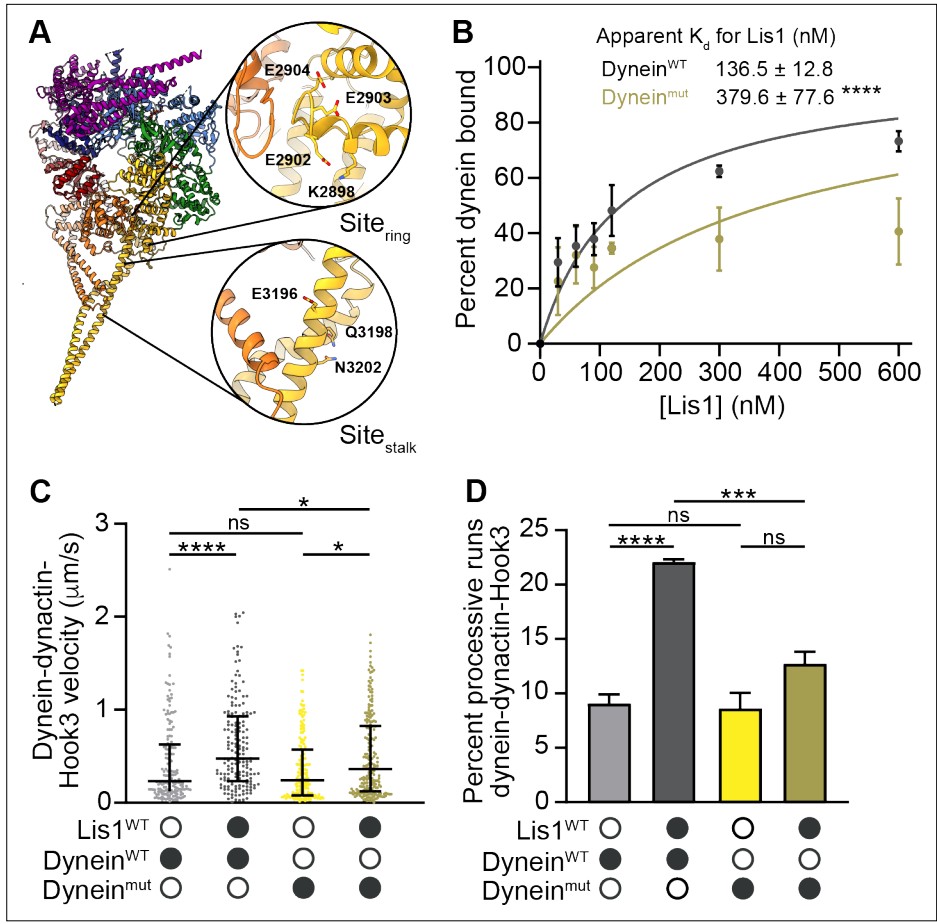

**Figure 8.** site$_{ring}$ and site$_{stalk}$ are important for Lis1's ability to increase dynein velocity. (**A**) Structure of the human dynein motor domain (PDB: 5NUG) (***Zhang et al., 2017***) highlighting residues mutated at site$_{ring}$ and site$_{stalk}$. (**B**) Quantitation of the binding affinity of human Lis1 for dynein$^{WT}$ (dark gray; K$_d$ = 136.5 ± 12.8) and Dynein$^{mut}$ (dark yellow; K$_d$ = 379.6 ± 77.6). n = 3 replicates per condition. Statistical analysis was performed using an extra sum-of-squares *F*-test; p=0.0001. (**C**) Single-molecule velocity of human dynein–dynactin–Hook3 complexes formed with dynein$^{WT}$ (shades of gray) or dynein$^{mut}$ (shades of yellow) in the absence or presence of 300 nM Lis1. Black circles indicate presence of a component in the assay, while white circles indicate the absence of a component. The median and interquartile range are shown. Statistical analysis was performed with a Kruskal–Wallis test; ****p<0.0001; *dynein$^{mut}$ in the presence and absence of Lis1 p=0.0190; *dynein$^{WT}$ and dynein$^{mut}$ in the presence of Lis1, p=0.0231; ns, p>0.9999. At least 150 single-molecule events were measured per condition. (**D**) Percentage (mean ± s.e.m.) of processive runs of dynein–dynactin–Hook3 complexes formed with dynein$^{WT}$ (shades of gray) or dynein$^{mut}$ (shades of yellow) in the absence or presence of 300 nM Lis1. n = 3 replicates per condition with each replicate including at least 200 individual single-molecule events. Black circles indicate presence of a component in the assay, while white circles indicate the absence of a component. Statistical analysis was performed with an ANOVA; ****p<0.0001; ***p=0.0008; ns, dynein$^{mut}$ in the presence and absence of Lis1, p=0.0755; ns, dynein$^{WT}$ and dynein$^{mut}$ in the absence of Lis1, p=0.9883.

The online version of this article includes the following figure supplement(s) for figure 8:

**Figure supplement 1.** Mutations in Lis1 reduce Lis1's ability to increase dynein's velocity.

(***Tarricone et al., 2004***). Here, we report the first high-resolution structure of yeast cytoplasmic dynein-1$^{E2488Q}$ bound to Lis1, which allowed us to build an atomic model (***Figure 1***). The structure revealed the details of the interactions between the Lis1 ß-propellers and its two binding sites on dynein, as well as how the two ß-propellers interact with each other. Our earlier work had identified the main binding sites on yeast dynein (***DeSantis et al., 2017***), but not their molecular details or their counterparts on Lis1. Our previous work with wild-type human dynein and Lis1 showed that Lis1 binds to dynein at similar sites in both systems (***Htet et al., 2020***). The much higher resolution of our current structure also revealed additional contacts between Lis1 and dynein that were not apparent in previous

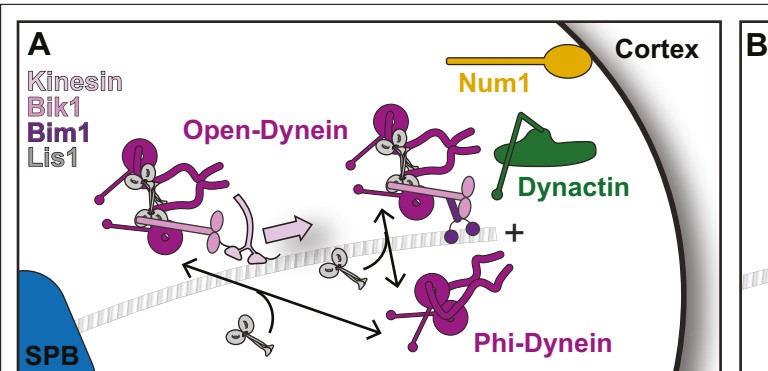
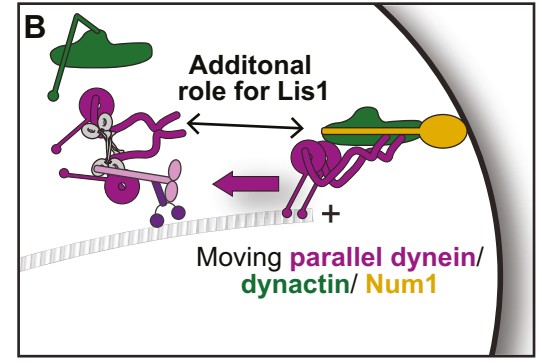

**Figure 9.** Model for Lis1 regulation of yeast dynein. (**A**) Model of dynein regulation by Lis1 in *S. cerevisiae*. In *S. cerevisiae*, dynein arrives at microtubule plus ends either via transport by kinesin in a complex that contains Lis1 or by recruitment from the cytoplasm. Dynein that is recruited from the cytoplasm is most likely in a Phi conformation. Lis1 binding to dynein, either in the kinesin transport complex or at microtubule plus ends, would favor an open conformation of dynein. (**B**) Our mutant analysis suggests that Lis1 has an additional role in dynein regulation after it disrupts Phi autoinhibition by binding to dynein at site$_{ring}$. Ultimately, dynein interacts with the candidate activating adaptor, Num1, on the cortex and Lis1 is released. We have opted to draw only a single dynein dimer in this complex as it is not known if these complexes contain one or two dynein dimers in yeast.

cryo-EM maps; these involve interactions at site$_{ring}$ between Lis1 and a loop in dynein's AAA5 domain (*Figure 1E*) and at site$_{stalk}$ between Lis1 and a loop in dynein's AAA4 domain (*Figure 1F*). Finally, being able to visualize the interface between the two Lis1 ß-propellers (*Figure 1G*) allowed us to test its role in the regulation of dynein by Lis1. Critical to achieving high resolution was the use of specialized streptavidin affinity grids (*Han et al., 2016*; *Han et al., 2012*; *Lahiri et al., 2019*), which helped overcome the preferred orientation of the sample, a major resolution-limiting factor in cryo-EM.

We used our new structure to guide mutagenesis of both yeast and human proteins to determine if the new contact sites we identified were important for dynein regulation by Lis1 in vitro and in vivo. Mutation of these sites at the endogenous alleles of yeast dynein and Lis1 revealed that they are all important for dynein and Lis1's in vivo function in mitotic spindle positioning, which ultimately leads to nuclear segregation following mitosis. We also found that Lis1 binding to dynein at site$_{ring}$ and site$_{stalk}$, and the Lis1–Lis1 interface were required for dynein to reach the cell cortex, the site where active yeast dynein–dynactin complexes interact with the candidate activating adaptor, Num1 (*Figure 9*; *Heil-Chapdelaine et al., 2000*; *Lammers and Markus, 2015*; *Lee et al., 2003*; *Sheeman et al., 2003*).

Our work also shows that the interaction of dynein and Lis1 at both site$_{ring}$ and site$_{stalk}$ is important for the formation of active human dynein–dynactin–activating adaptor complexes containing either the Hook3 or BicD2-activating adaptors. Previously, we and others found that Lis1 was also important for the formation of activated dynein complexes containing the activating adaptors Ninl and BicDL1 (also called BicDR1) (*Elshenawy et al., 2020*; *Htet et al., 2020*). More recent work by us and others showed that the activating adaptors TRAK2 and Hook2 are almost completely reliant on Lis1 for complex formation (*Christensen et al., 2021*; *Fenton et al., 2021*). Thus, it is likely that this role for Lis1 in complex assembly, and Lis1 binding to dynein at site$_{ring}$ and site$_{stalk}$, will be required for dynein complex formation driven by other activating adaptors. The importance of site$_{ring}$ and site$_{stalk}$ in both human and yeast dynein activation supports a conserved mode of Lis1 regulation of dynein.

How does Lis1 promote dynein–dynactin–activating adaptor complex assembly? Our structure of a dynein$^{E2488Q}$–(Lis1)$_2$ complex shows that binding of Lis1 to dynein at site$_{ring}$ is sterically incompatible with formation of the autoinhibited Phi conformation of dynein (*Zhang et al., 2017*). Based on our lower resolution structures (*DeSantis et al., 2017*), we and others (*Canty and Yildiz, 2020*; *Elshenawy et al., 2020*; *Htet et al., 2020*; *Markus et al., 2020*; *Marzo et al., 2020*; *Qiu et al., 2019*; *Xiang and Qiu, 2020*) had proposed that Lis1 binding to dynein at site$_{ring}$ may shift the equilibrium from the Phi to an open conformation of dynein, ultimately allowing dynein complexes to assemble with dynactin and an activating adaptor (*Figure 9*). In yeast, dynein can be recruited to microtubule plus ends either via transport by a kinesin, Kip2 (*Carvalho et al., 2004*; *Caudron et al., 2008*; *Markus et al., 2009*), or by recruitment from the cytosol (*Caudron et al., 2008*; *Markus et al., 2009*). We propose that cytosolic dynein is in a Phi conformation, while dynein in the kinesin transport complex or at microtubule plus ends is no longer in its autoinhibited state (*Figure 9A*). The kinesin transport

complex contains Lis1 (*Roberts et al., 2014*), and Lis1 is also required for dynein's plus end localization (*Lee et al., 2003*; *Sheeman et al., 2003*).

Does Lis1 play additional roles in dynein complex assembly beyond disrupting dynein's autoinhibited conformation? In our nuclear segregation assays, Phi disrupting mutants in yeast dynein completely rescued the nuclear segregation defects seen with a $site_{ring}$ Lis1 mutation (*Figure 7*). However, in more sensitive in vivo assays in both *Aspergillus* and yeast, Phi disrupting dynein mutants did not completely rescue Lis1 mutations or deletion (*Marzo et al., 2020*; *Qiu et al., 2019*). In addition, we showed previously that the presence of Lis1 increased the percentage of dynein–dynactin–activating adaptor complexes that contained two dynein dimers using human proteins in vitro (*Htet et al., 2020*). Using a Phi-disrupting dynein mutant in these assays with human proteins decreased, but did not completely abolish the requirement for Lis1 (*Htet et al., 2020*). Together, these experiments suggest that Lis1 may have additional roles in dynein complex assembly beyond disrupting dynein's autoinhibited Phi conformation. In support of this, we showed here that a Phi-disrupting mutation in dynein was less efficient in reaching the cell cortex (where dynein is active) when Lis1 mutated at $site_{ring}$ was present in the same genetic background (*Figure 7*). This suggests that the contact between dynein and Lis1 at $site_{ring}$ has an additional role in regulating dynein downstream of autoinhibition relief (*Figure 9B*). This additional role of Lis1 could be promoting further conformational changes in dynein. Our data suggest that there could be a role for Lis1's modulation of dynein's microtubule binding affinity in this process as mutations at $site_{ring}$ (which normally promotes tight microtubule binding) lead to less Phi-disrupted dynein reaching the cortex. While dynein's microtubule binding domain is not required to reach the cortex (*Lammers and Markus, 2015*), our data suggest some role for microtubule binding in this process. Assembling active dynein–dynactin–activating adaptor complexes likely involves multiple steps, and the many in vitro phenotypes described for Lis1's regulation of dynein, such as the nucleotide-specific affects at AAA3 (*DeSantis et al., 2017*; *Qiu et al., 2021*), may be providing mechanistic hints about these steps. This will be an important area to investigate in the future.

Overall, this work and that of others suggests that the field is coalescing around a unified model for dynein regulation of Lis1 across species (*Canty and Yildiz, 2020*; *Elshenawy et al., 2020*; *Htet et al., 2020*; *Markus et al., 2020*; *Marzo et al., 2020*; *Qiu et al., 2019*; *Xiang and Qiu, 2020*).

## Materials and methods

### Yeast strain construction

The *S. cerevisiae* strains used in this study are listed in *Supplementary file 1*. The endogenous genomic copies of *DYN1* and *PAC1* (encoding the dynein heavy chain and Lis1, respectively) were modified or deleted using PCR-based methods as previously described (*Longtine et al., 1998*). Transformations were performed using the lithium acetate method and screened by colony PCR. Point mutants were generated using QuikChange site-directed mutagenesis (Agilent) and verified by DNA sequencing.

### Nuclear segregation assay

Log-phase *S. cerevisiae* cells growing at 30°C were transferred to 16°C for 16 hr. Cells were fixed with 75% ethanol and stained with DAPI. Imaging was performed using a ×100 Apo TIRF NA 1.49 objective on a Nikon Ti2 microscope with a Yokogawa-X1 spinning disk confocal system, MLC400B laser engine (Agilent), Prime 95B back-thinned sCMOS camera (Teledyne Photometrics), and a piezo Z-stage (Mad City Labs). The percentage of aberrant binucleate cells was calculated as the number of binucleate cells divided by the sum of wild-type and binucleate cells. Six (*Figures 3C–5B*), four (*Figure 7A*), or three (*Figure 3D*) biological replicates were performed from independent cultures for each condition.

### Live-cell imaging

Mid log-phase *S. cerevisiae* cells were mounted on an agarose pad made from SC media. For *Figures 6 and 7*, live cells endogenously modified to express fluorescently labeled *DYN1-3XGFP*, *CFP-TUB1*, and *SPC110-tdTomato* were imaged using a Yokogawa W1 confocal scanhead mounted to a Nikon Ti2 microscope with an Apo TIRF 100 × 1.49 NA objective. The microscope was run with NIS Elements using the 488 nm and 561 nm lines of a six-line (405 nm, 445 nm, 488 nm, 515 nm, 561 nm, and 640 nm) LUN-F-XL laser engine and Prime95B cameras (Photometrics). The localization

of *DYN1-3XGFP* foci was assessed and the number of foci on each location (SPB, microtubule plus end, and cell cortex) was counted.

## Cloning, plasmid construction, and mutagenesis

The pDyn1 plasmid (the pACEBac1 expression vector containing insect cell codon-optimized dynein heavy chain [*DYNC1H1*] fused to a His-ZZ-TEV tag on the amino-terminus and a carboxy-terminal SNAPf tag [New England Biolabs]) and the pDyn2 plasmid (the pIDC expression vector with codon optimized *DYNC1I2*, *DYNC1LI2*, *DYNLT1*, *DYNLL1*, and *DYNLRB1*) were recombined in vitro with a Cre recombinase (New England Biolabs) to generate the pDyn3 plasmid. The presence of all six dynein chains was verified by PCR. pDyn1, pDyn2, and the pFastBac plasmid with codon-optimized human full-length Lis1 (*PAFAH1B1*) fused to an amino-terminal His-ZZ-TEV tag was a gift from Andrew Carter (LMB-MRC, Cambridge, UK). The Hook3 construct contained amino acids 1–552 and was obtained as described previously (*Redwine et al., 2017*). Activating adaptors were fused to a ZZ-TEV-HaloTag (Promega) on the amino-terminus and inserted into a pET28a expression vector. All additional tags were added via Gibson assembly, and all mutations and truncations were made via site-directed mutagenesis (Agilent).

## *S. cerevisiae* immunoprecipitations and western blots

Log-phase *S. cerevisiae* cells grown at 30°C were pelleted at 4000 × *g* and frozen using liquid nitrogen. Liquid nitrogen-frozen yeast cell pellets were lysed by grinding in a chilled coffee grinder and resuspended in DLB (30 mM HEPES [pH 7.4], 50 mM potassium acetate, 2 mM magnesium acetate, 1 mM EGTA, 10% glycerol, 1 mM DTT) supplemented with 1 mM Pefabloc, 0.2% Triton, cOmplete EDTA-free protease inhibitor cocktail tablet (Roche), and 1 mM Pepstatin A (Cayman Chemical Company). The lysate was clarified by centrifuging at 50,000 × *g* for 1 hr. The protein concentration of the clarified supernatants was quantified using a Bradford Protein Assay (Bio-Rad), and equal amounts of clarified lysates were incubated with either ANTI-FLAG M2 Affinity Gel (Sigma) or Anti-GFP nanobody beads (made in house) for 2 hr at 4°C. Beads were washed with DLB buffer, boiled in SDS sample buffer and loaded onto an NuPAGE Bis-Tris gel (Invitrogen). Gels were transferred to a PVDF membrane that was blocked with PBS-T (PBS1X and 0.1% Tween-20) containing 5% milk and 1% BSA for 1 hr at room temperature and blotted with either a rabbit anti-FLAG antibody (1:3000; Proteintech 20543-1-AP) or a mouse anti-GFP antibody (1:3000; Santa Cruz sc-9996) overnight at 4°C. Membranes were then incubated with goat-anti rabbit IRDye 800RD or goat anti-mouse IRDye 680RD secondary antibodies (LI-COR), respectively, and were scanned with a LI-COR Odyssey imaging system.

## *S. cerevisiae* protein purification

Protein purification steps were done at 4°C unless otherwise indicated. Dynein constructs were purified from *S. cerevisiae* using a ZZ tag as previously described (*Reck-Peterson et al., 2006*). Briefly, liquid nitrogen-frozen yeast cell pellets were lysed by grinding in a chilled coffee grinder and resuspended in dynein lysis buffer (DLB: 30 mM HEPES [pH 7.4], 50 mM potassium acetate, 2 mM magnesium acetate, 1 mM EGTA, 10% glycerol, 1 mM DTT) supplemented with 0.1 mM Mg-ATP, 0.5 mM Pefabloc, 0.05% Triton, and cOmplete EDTA-free protease inhibitor cocktail tablet (Roche). The lysate was clarified by centrifuging at 264,900 × *g* for 1 hr or at 125,100 × *g* for 2 hr. The clarified supernatant was incubated with IgG Sepharose beads (GE Healthcare Life Sciences) for 1 hr. The beads were transferred to a gravity flow column, washed with DLB buffer supplemented with 250 mM potassium chloride, 0.1 mM Mg-ATP, 0.5 mM Pefabloc and 0.1% Triton, and with TEV buffer (10 mM Tris–HCl [pH 8.0], 150 mM potassium chloride, 10% glycerol, 1 mM DTT, and 0.1 mM Mg-ATP). GST-dimerized dynein constructs were labeled with 5 mM Halo-TMR (Promega) in the column for 10 min at room temperature, and the unbound dye was washed away with TEV buffer at 4°C. Dynein was cleaved from IgG beads via incubation with 0.15 mg/mL TEV protease (purified in the Reck-Peterson lab) for 1 hr at 16°C. For dynein monomer constructs, the TEV cleavage step was done overnight at 4°C and the cleaved proteins were concentrated using 100K MWCO concentrator (EMD Millipore). Cleaved proteins were filtered by centrifuging with Ultrafree-MC VV filter (EMD Millipore) in a tabletop centrifuge and flash frozen in liquid nitrogen.

Lis1 constructs were purified from *S. cerevisiae* using their His$_8$ and ZZ tags as previously described (*Huang et al., 2012*). Lysis and clarification steps were similar to dynein purification except for the lysis

buffer used was buffer A (50 mM potassium phosphate [pH 8.0], 150 mM potassium acetate, 150 mM sodium chloride, 2 mM magnesium acetate, 5 mM β-mercaptoethanol, 10% glycerol, 0.2% Triton, 0.5 mM Pefabloc) supplemented with 10 mM imidazole (pH 8.0) and cOmplete EDTA-free protease inhibitor cocktail tablet. The clarified supernatant was incubated with Ni-NTA agarose (QIAGEN) for 1 hr. The Ni beads were transferred to a gravity column, washed with buffer A + 20 mM imidazole (pH 8.0), and eluted with buffer A + 250 mM imidazole (pH 8.0). The eluted protein was incubated with IgG Sepharose beads for 1 hr. IgG beads were transferred to a gravity flow column, washed with buffer A + 20 mM imidazole (pH 8.0) and with modified TEV buffer (50 mM Tris–HCl [pH 8.0], 150 mM potassium acetate, 2 mM magnesium acetate, 1 mM EGTA, 10% glycerol, 1 mM DTT). Lis1 was cleaved from the IgG beads via incubation with 0.15 mg/mL TEV protease (purified in the Reck-Peterson lab) for 1 hr at 16°C. Cleaved proteins were filtered by centrifuging with Ultrafree-MC VV filter (EMD Millipore) in a tabletop centrifuge and flash frozen in liquid nitrogen.

## Human protein purification

Human full-length dynein, human dynein monomer, and human Lis1 constructs were expressed in Sf9 cells as described previously (*Baumbach et al., 2017*; *Htet et al., 2020*; *Schlager et al., 2014*). Briefly, the pDyn3 plasmid containing the human dynein genes or the pFastBac plasmid containing full-length Lis1 was transformed into DH10EmBacY chemically competent cells with heat shock at 42°C for 15 s followed by incubation at 37°C for 5 hr in S.O.C media (Thermo Fisher Scientific). The cells were then plated on LB-agar plates containing kanamycin (50 µg/mL), gentamicin (7 µg/mL), tetracycline (10 µg/mL), BluoGal (100 µg/mL), and IPTG (40 µg/mL), and positive clones were identified by a blue/white color screen after 48 hr. For full-length human dynein constructs, white colonies were additionally tested for the presence of all six dynein genes using PCR. These colonies were then grown overnight in LB medium containing kanamycin (50 µg/mL), gentamicin (7 µg/mL), and tetracycline (10 µg/mL) at 37°C. Bacmid DNA was extracted from overnight cultures using an isopropanol precipitation method as described previously (*Zhang et al., 2017*). 2 mL of Sf9 cells at $0.5 \times 10^6$ cells/mL were transfected with 2 µg of fresh bacmid DNA and FuGene HD transfection reagent (Promega) at a 3:1 transfection reagent to DNA ratio according to the manufacturer's instructions. After 3 days, the supernatant containing the 'V0' virus was harvested by centrifugation at $200 \times g$ for 5 min at 4°C. To generate 'V1,' 1 mL of the V0 virus was used to transfect 50 mL of Sf9 cells at $1 \times 10^6$ cells/mL. After 3 days, the supernatant containing the V1 virus was harvested by centrifugation at $200 \times g$ for 5 min at 4°C and stored in the dark at 4°C until use. For protein expression, 4 mL of the V1 virus were used to transfect 400 mL of Sf9 cells at $1 \times 10^6$ cells/mL. After 3 days, the cells were harvested by centrifugation at $3000 \times g$ for 10 min at 4°C. The pellet was resuspended in 10 mL of ice-cold PBS and pelleted again. The pellet was flash frozen in liquid nitrogen and stored at –80°C.

Protein purification steps were done at 4°C unless otherwise indicated. Full-length dynein and dynein monomer were purified from frozen Sf9 pellets transfected with the V1 virus as described previously (*Schlager et al., 2014*). Frozen cell pellets from a 400 mL culture were resuspended in 40 mL of Dynein-lysis buffer (50 mM HEPES [pH 7.4], 100 mM sodium chloride, 1 mM DTT, 0.1 mM Mg-ATP, 0.5 mM Pefabloc, 10% [v/v] glycerol) supplemented with 1 cOmplete EDTA-free protease inhibitor cocktail tablet (Roche) per 50 mL and lysed using a Dounce homogenizer (10 strokes with a loose plunger and 15 strokes with a tight plunger). The lysate was clarified by centrifuging at $183,960 \times g$ for 88 min in Type 70 Ti rotor (Beckman). The clarified supernatant was incubated with 4 mL of IgG Sepharose 6 Fast Flow beads (GE Healthcare Life Sciences) for 3–4 hr on a roller. The beads were transferred to a gravity flow column, washed with 200 mL of Dynein-lysis buffer and 300 mL of TEV buffer (50 mM Tris–HCl [pH 8.0], 250 mM potassium acetate, 2 mM magnesium acetate, 1 mM EGTA, 1 mM DTT, 0.1 mM Mg-ATP, 10% [v/v] glycerol). For fluorescent labeling of carboxy-terminal SNAPf tag, dynein-coated beads were labeled with 5 µM SNAP-Cell-TMR (New England Biolabs) in the column for 10 min at room temperature and unbound dye was removed with a 300 mL wash with TEV buffer at 4°C. The beads were then resuspended and incubated in 15 mL of TEV buffer supplemented with 0.5 mM Pefabloc and 0.2 mg/mL TEV protease (purified in the Reck-Peterson lab) overnight on a roller. The supernatant containing cleaved proteins was concentrated using a 100K MWCO concentrator (EMD Millipore) to 500 µL and purified via size-exclusion chromatography on a TSKgel G4000SWXL column (Tosoh Bioscience) with GF150 buffer (25 mM HEPES [pH7.4], 150 mM potassium chloride, 1 mM magnesium chloride, 5 mM DTT, 0.1 mM Mg-ATP) at 1 mL/min. The peak

fractions were collected, buffer exchanged into a GF150 buffer supplemented with 10% glycerol, concentrated to 0.1–0.5 mg/mL using a 100K MWCO concentrator (EMD Millipore), and flash frozen in liquid nitrogen.

Lis1 constructs were purified from frozen cell pellets from 400 mL culture. Lysis and clarification steps were similar to full-length dynein purification except Lis1-lysis buffer (30 mM HEPES [pH 7.4], 50 mM potassium acetate, 2 mM magnesium acetate, 1 mM EGTA, 300 mM potassium chloride, 1 mM DTT, 0.5 mM Pefabloc, 10% [v/v] glycerol) supplemented with 1 cOmplete EDTA-free protease inhibitor cocktail tablet (Roche) per 50 mL was used. The clarified supernatant was incubated with 0.5 mL of IgG Sepharose 6 Fast Flow beads (GE Healthcare Life Sciences) for 2–3 hr on a roller. The beads were transferred to a gravity flow column, washed with 20 mL of Lis1-lysis buffer, 100 mL of modified TEV buffer (10 mM Tris–HCl [pH 8.0], 2 mM magnesium acetate, 150 mM potassium acetate, 1 mM EGTA, 1 mM DTT, 10% [v/v] glycerol) supplemented with 100 mM potassium acetate, and 50 mL of modified TEV buffer. For fluorescent labeling of Lis1 constructs with amino-terminal HaloTags, Lis1-coated beads were labeled with 200 µM Halo-TMR (Promega) for 2.5 hr at 4°C on a roller and the unbound dye was removed with a 200 mL wash with modified TEV buffer supplemented with 250 mM potassium acetate. Lis1 was cleaved from IgG beads via incubation with 0.2 mg/mL TEV protease overnight on a roller. The cleaved Lis1 was filtered by centrifuging with an Ultrafree-MC VV filter (EMD Millipore) in a tabletop centrifuge and flash frozen in liquid nitrogen.

Dynactin was purified from stable HEK293T cell lines expressing p62-Halo-3xFlag as described previously (*Redwine et al., 2017*). Briefly, frozen pellets collected from 160 × 15 cm plates were resuspended in 80 mL of Dynactin-lysis buffer (30 mM HEPES [pH 7.4], 50 mM potassium acetate, 2 mM magnesium acetate, 1 mM EGTA, 1 mM DTT, 10% [v/v] glycerol) supplemented with 0.5 mM Mg-ATP, 0.2% Triton X-100, and 1 cOmplete EDTA-free protease inhibitor cocktail tablet (Roche) per 50 mL and rotated slowly for 15 min. The lysate was clarified by centrifuging at 66,000 × *g* for 30 min in Type 70 Ti rotor (Beckman). The clarified supernatant was incubated with 1.5 mL of anti-Flag M2 affinity gel (Sigma-Aldrich) overnight on a roller. The beads were transferred to a gravity flow column, washed with 50 mL of wash buffer (Dynactin-lysis buffer supplemented with 0.1 mM Mg-ATP, 0.5 mM Pefabloc, and 0.02% Triton X-100), 100 mL of wash buffer supplemented with 250 mM potassium acetate, and again with 100 mL of wash buffer. For fluorescent labeling, the HaloTag, dynactin-coated beads were labeled with 5 µM Halo-JF646 (Janelia) in the column for 10 min at room temperature and the unbound dye was washed with 100 mL of wash buffer at 4°C. Dynactin was eluted from beads with 1 mL of elution buffer (wash buffer with 2 mg/mL of 3xFlag peptide). The eluate was collected, filtered by centrifuging with Ultrafree-MC VV filter (EMD Millipore) in a tabletop centrifuge, and diluted to 2 mL in buffer A (50 mM Tris-HCl [pH 8.0], 2 mM magnesium acetate, 1 mM EGTA, and 1 mM DTT) and injected onto a MonoQ 5/50 GL column (GE Healthcare and Life Sciences) at 1 mL/min. The column was pre-washed with 10 CV of buffer A, 10 CV of buffer B (50 mM Tris-HCl [pH 8.0], 2 mM magnesium acetate, 1 mM EGTA, 1 mM DTT, 1 M potassium acetate), and again with 10 CV of buffer A at 1 mL/min. To elute, a linear gradient was run over 26 CV from 35 to 100% buffer B. Pure dynactin complex eluted from ~75 to 80% buffer B. Peak fractions containing pure dynactin complex were pooled, buffer exchanged into a GF150 buffer supplemented with 10% glycerol, concentrated to 0.02–0.1 mg/mL using a 100K MWCO concentrator (EMD Millipore), and flash frozen in liquid nitrogen.

Activating adaptors containing amino-terminal HaloTags were expressed in BL-21[DE3] cells (New England Biolabs) at OD 0.4–0.6 with 0.1 mM IPTG for 16 hr at 18°C. Frozen cell pellets from 2 L culture were resuspended in 60 mL of activator-lysis buffer (30 mM HEPES [pH 7.4], 50 mM potassium acetate, 2 mM magnesium acetate, 1 mM EGTA, 1 mM DTT, 0.5 mM Pefabloc, 10% [v/v] glycerol) supplemented with 1 cOmplete EDTA-free protease inhibitor cocktail tablet (Roche) per 50 mL and 1 mg/mL lysozyme. The resuspension was incubated on ice for 30 min and lysed by sonication. The lysate was clarified by centrifuging at 66,000 × *g* for 30 min in Type 70 Ti rotor (Beckman). The clarified supernatant was incubated with 2 mL of IgG Sepharose 6 Fast Flow beads (GE Healthcare Life Sciences) for 2 hr on a roller. The beads were transferred to a gravity flow column, washed with 100 mL of activator-lysis buffer supplemented with 150 mM potassium acetate and 50 mL of cleavage buffer (50 mM Tris–HCl [pH 8.0], 150 mM potassium acetate, 2 mM magnesium acetate, 1 mM EGTA, 1 mM DTT, 0.5 mM Pefabloc, 10% [v/v] glycerol). The beads were then resuspended and incubated in 15 mL of cleavage buffer supplemented with 0.2 mg/mL TEV protease overnight on a roller. The supernatant containing cleaved proteins was concentrated using a 50K MWCO concentrator (EMD

Millipore) to 1 mL, filtered by centrifuging with Ultrafree-MC VV filter (EMD Millipore) in a tabletop centrifuge, diluted to 2 mL in buffer A (30 mM HEPES [pH 7.4], 50 mM potassium acetate, 2 mM magnesium acetate, 1 mM EGTA, 10% [v/v] glycerol, and 1 mM DTT) and injected onto a MonoQ 5/50 GL column (GE Healthcare and Life Sciences) at 1 mL/min. The column was pre-washed with 10 CV of buffer A, 10 CV of buffer B (30 mM HEPES [pH 7.4], 1 M potassium acetate, 2 mM magnesium acetate, 1 mM EGTA, 10% [v/v] glycerol, and 1 mM DTT) and again with 10 CV of buffer A at 1 mL/min. To elute, a linear gradient was run over 26 CV from 0 to 100% buffer B. The peak fractions containing Halo-tagged activating adaptors were collected and concentrated using a 50K MWCO concentrator (EMD Millipore) to 0.2 mL. For fluorescent labeling the HaloTag, the concentrated peak fractions were incubated with 5 µM Halo-Alexa488 (Promega) for 10 min at room temperature. Unbound dye was removed by PD-10 desalting column (GE Healthcare and Life Sciences) according to the manufacturer's instructions. The labeled activating adaptor sample was concentrated using a 50K MWCO concentrator (EMD Millipore) to 0.2 mL, diluted to 0.5 mL in GF150 buffer, and further purified via size-exclusion chromatography on a Superose 6 Increase 10/300 GL column (GE Healthcare and Life Sciences) with GF150 buffer at 0.5 mL/min. The peak fractions were collected, buffer exchanged into a GF150 buffer supplemented with 10% glycerol, concentrated to 0.2–1 mg/mL using a 50K MWCO concentrator (EMD Millipore), and flash frozen in liquid nitrogen.

## Electron microscopy sample preparation

Monomeric yeast dynein$^{E2488Q}$ was buffer exchanged into Modification Buffer (100 mM sodium phosphate, pH 8.0, 150 mM sodium chloride) and biotinylated using water-soluble Sulfo ChromaLink biotin (Sulfo ChromaLink Biotin, Cat# B-1007) in a 1:2 molar ratio for 2 hr at room temperature. Biotinylated protein was dialyzed into TEV buffer overnight at 4°C. Biotinylation of dynein$^{E2488Q}$ was verified using a pull-down assay with streptavidin magnetic beads (Thermo Fisher).

Streptavidin affinity grids (SA grids) were prepared as previously described (*Han et al., 2016*). Just prior to use, the SA grids were washed by touching the SA side of the grid to three drops (2 × 50 µL, 1 × 100 µL) of rehydration buffer (150 mM potassium chloride, 50 mM HEPES pH 7.4, 5 mM EDTA) to remove the storage trehalose layer. For complete removal of the protective trehalose layer, rehydration buffer was manually pipetted up and down onto the SA side of the grid, and then the grid was placed onto a 100 µL drop of rehydration buffer for 10 min. This process was repeated once, and then the grid was buffered exchanged into sample buffer using five 50 µL drops of sample buffer (TEV buffer without glycerol and DTT, supplemented with 1.2 mM ATP and 1.2 mM Na$_3$VO$_4$). 4 µL of sample (150 nM dynein$^{E2488Q}$, 650 nM Lis1, 1.2 mM ATP, 1.2 mM Na$_3$VO$_4$) was applied to each grid and incubated for ~10 min inside a humidity chamber. Unbound protein was washed away by touching the sample side of the grid to three 100 µL drops of wash buffer supplemented with 150 nM dimeric yeast Lis1 (26 mM Tris pH 8, 75 mM potassium chloride, 64 mM potassium acetate, 0.85 mM magnesium acetate, 0.5 mM EGTA, 1.2 mM ATP, 1.2 mM Na$_3$VO$_4$). Grids were vitrified using a Vitrobot (FEI) set to 20°C and 100% humidity. Vitrobot tweezers were kept cold on ice between samples. The sample was first manually wicked inside the Vitrobot using a Whatman No. 1 filter paper, followed by the addition of 1.5 µL of sample buffer before standard blotting and vitrification proceeded (blot time: 4 s; blot force: 20).

## Image collection and processing

Vitrified grids were imaged on a Titan Krios (FEI) operated at an accelerating voltage of 300 kV, and the images were recorded with a K2 Summit direct electron detector (Gatan Inc). We collected 2378 movies in super-resolution mode (0.655 Å/super-resolution pixel at the object level) dose-fractionated into 200 ms frames for a total exposure of 10 s with a dose rate of 10 electrons/pixel/s for a total fluence of 58.3 electrons/Å$^2$. The defocus of the images varied within a range of –2.0 µm to –2.7 µm. Automated data collection was executed by SerialEM (*Mastronarde, 2005*).

Video frames were aligned using the dose-weighted frame alignment option in UCSF MotionCor2 (*Zheng et al., 2017*) as employed in Relion 3.0 (*Zivanov et al., 2018*). At this stage, the individual frames were corrected for the anisotropic magnification distortion inherent to the electron microscope (*Grant and Grigorieff, 2015*). The signal from the streptavidin lattice was removed from aligned micrographs using Fourier filtering as described previously (*Han et al., 2012*). Micrographs were manually inspected for defects including suboptimal ice thickness and incomplete removal of the SA

lattice signal, and 46 micrographs were removed leaving 2332 micrographs for further processing. CTF estimation was carried out on the non-dose-weighted micrographs using GCTF (*Zhang, 2016*) using the local CTF estimation option as implemented in Appion (*Lander et al., 2009*). Images with CTF fits having 0.5 confidence resolution worse than 5 Å were excluded from further processing. Particles were picked using crYOLO (*Wagner et al., 2019*) using a training model generated from manually picked particles. The particles were extracted with a downsampled pixel size of 3.93 Å/pixel, and a single round of two-dimensional (2D) classification was carried out in Relion 3.0 to identify bad particles. Particles belonging to good 2D class averages were recentered and extracted (1.31 Å/pixel), and further processed in cryoSPARC. Ab initio models were generated with the majority of particles going into one good class displaying the characteristic features of the dynein motor domain and the WD40 rings of Lis1. Those particles were refined using the non-uniform refinement routine of cryoSPARC (*Punjani et al., 2020*), against the best ab initio model. This resulted in a 3.2 Å map containing 233,476 particles. We performed per-particle CTF refinement and beam tilt refinement in Relion 3.0 followed by a single round of three-dimensional (3D) classification without alignment. The particles contributing to one of the classes lead to a high-resolution reconstruction of the dynein–Lis1 complex, and those particles were further refined in Relion 3.0 to generate the final 3.1 Å map (based on the 0.143 cutoff of the gold-standard Fourier shell correlation [FSC] curve) of the dynein$^{E2488Q}$-Lis1 complex. For visualization purposes, the map was sharpened with an automatically estimated negative B-factor of 33 (as determined from the 'PostProcess' routine of Relion 3.0). The local resolution of the map was estimated using the 'local resolution' routine of Relion 3.0, and the map was low-pass filtered according to the local resolution prior to analysis. The 3-D FSC was calculated by 3dfsc version 2.5 (*Tan et al., 2017*) using the half-maps and the mask used for the PostProcess routine in Relion 3.0.

## Model building

The yeast dynein$^{E2488Q}$-Lis1 map was segmented using Seggar (*Pintilie et al., 2010*) as implemented in UCSF chimera (*Pettersen et al., 2004*), and the atomic models of the different components were initially built to account for their respective segmented maps. A homology of the Lis1 WD40 domain was generated using I-TASSER (*Roy et al., 2010*), and this model was rigid body docked into the corresponding segmented map using the 'fit in map' routing in UCSF Chimera. Regions of the model that did not agree with the EM map were manually rebuilt using Coot (*Emsley et al., 2010*). The resulting model was used as a reference for Rosetta CM (*Song et al., 2013*) and 1200 models were generated. A hybrid model was made using the two lowest energy output models from Rosetta CM. This model was then manually placed using Coot into both Lis1 sites, and areas of disagreement between the map and the model were resolved manually.

A homology model of the yeast dynein motor domain was created in Swiss-Model (*Waterhouse et al., 2018*) by using the human dynein atomic model in the closed conformation (PDB: 5NUG) (*Zhang et al., 2017*) as a template. The resulting model was fit in the segmented map corresponding to the dynein motor domain using the 'fit to map' command in UCSF Chimera and manually rebuilt in Coot to improve the agreement between the atomic model and the map. The resulting model was used as a template for Rosetta CM and 700 models were generated. The lowest energy model was selected and manually placed into the map, followed by an additional round of manual rebuilding.

The resulting models of the yeast dynein$^{E2488Q}$ motor domain and yeast Lis1 WD40 domains were combined and further refined against the unsegmented dynein$^{E2488Q}$-Lis1 map using an using an iterative process between Phenix Real Space Refine (*Afonine et al., 2018*) and manual rebuilding in Coot.

3D variable analysis (*Punjani et al., 2020*) was carried out in cryoSPARC using the particles and mask used in the non-uniform refinement job, resulting in the 3.2 Å map (*Figure 1—figure supplement 1*). Default parameters were used to solve three modes, and results were filtered to 5 Å. Results were visualized using the 3D variability display using the simple output mode with 20 frames/cluster, and then viewed using ChimeraX (*Pettersen et al., 2021*).

## Binding curves

To assess dynein/Lis1 binding, Lis1 was first covalently coupled to 16 µL of SNAP-Capture Magnetic Beads (New England Biolabs) in 2 mL Protein Lo Bind Tubes (Eppendorf) using the following protocol. Beads were washed twice with 1 mL of modified TEV buffer. Lis1 (0–600 nM) was added to the beads and gently shaken for 1 hr. The supernatant was analyzed via SDS-PAGE to confirm complete depletion

of Lis1. The Lis1-conjugated beads were washed once with 1 mL modified TEV buffer and once with 1 mL of binding buffer (10 mM Tris–HCl [pH 8.0], 150 mM potassium chloride, 2 mM magnesium chloride, 10% glycerol, 1 mM DTT, 0.1% NP40, 1 mM ADP). 20 nM wild-type dynein (monomeric motor domain) diluted in binding buffer was added to the Lis1-conjugated beads and gently agitated for 30 min. The supernatant was analyzed via SDS-PAGE, stained with SYPRO Red (Thermo Fisher), and dynein depletion was determined using densitometry in ImageJ. Binding curves were fit in Prism8 (GraphPad) with a nonlinear regression for one site binding with Bmax set to 1. Three technical replicates were collected per condition.

## Single-molecule TIRF microscopy

Single-molecule imaging was performed with an inverted microscope (Nikon, Ti-E Eclipse) equipped with a 100 × 1.49 NA oil immersion objective (Nikon, Plano Apo), a ProScan linear motor stage controller (Prior), and a LU-NV laser launch (Nikon), with 405 nm, 488 nm, 532 nm, 561 nm, and 640 nm laser lines. The excitation and emission paths were filtered using appropriate single band-pass filter cubes (Chroma). The emitted signals were detected with an electron multiplying CCD camera (Andor Technology, iXon Ultra 897). Illumination and image acquisition wee controlled by NIS Elements Advanced Research software (Nikon).

Single-molecule motility and microtubule binding assays were performed in flow chambers assembled as described previously (Case 1997) using the TIRF microscopy set-up described above. Either biotin-PEG-functionalized coverslips (Microsurfaces) or No. 1-1/2 coverslips (Corning) sonicated in 100% ethanol for 10 min were used for the flow-chamber assembly. Taxol-stabilized microtubules with ~10% biotin-tubulin and ~10% fluorescent-tubulin (Alexa405- or 488-labeled) were prepared as described previously (*Roberts et al., 2014*). Flow chambers were assembled with taxol-stabilized microtubules by incubating sequentially with the following solutions, interspersed with two washes with assay buffer (30 mM HEPES [pH 7.4], 2 mM magnesium acetate, 1 mM EGTA, 10% glycerol, 1 mM DTT) supplemented with 20 μM taxol and 50 mM potassium acetate in between (1) 1 mg/mL biotin-BSA in assay buffer (3 min incubation, ethanol washed coverslips only), (2) 1 mg/mL streptavidin in assay buffer (3 min incubation), and (3) a fresh dilution of taxol-stabilized microtubules in assay buffer (3 min incubation). After flowing in microtubules, the flow chamber was washed twice with assay buffer supplemented with 1 mg/mL casein and 20 μM taxol. Three technical replicates were collected per condition.

For assays using *S. cerevisiae* proteins, dynein was incubated with Lis1 or modified TEV buffer (to buffer match for Lis1) in assay buffer supplemented with 50 mM potassium acetate for 10 min before flowing into the assembled flow chamber. The final assay buffer was supplemented with 1 mg/mL casein, 71.5 mM β-mercaptoethanol, an oxygen scavenger system (0.4% glucose, 45 mg/mL glucose catalase, and 1.15 mg/mL glucose oxidase), and 2.5 mM Mg-ATP. For experiments in the presence of vanadate, 2.5 mM sodium vanadate was included. The final concentration of dynein was 2–15 pM. For single-molecule microtubule binding assays, the final imaging mixture containing dynein was incubated for an additional 5 min in the flow chamber at room temperature before imaging. After 5 min incubation, microtubules were imaged first by taking a single-frame snapshot. Dynein was imaged by taking a single-frame snapshot. Each sample was imaged at four different fields of view, and there were between 5 and 10 microtubules in each field of view. In order to compare the effect of Lis1 on microtubule binding, the samples with and without Lis1 were imaged in two separate flow chambers made on the same coverslip on the same day with the same stock of polymerized tubulin as described previously (*Roberts et al., 2014*). For single-molecule motility assays, microtubules were imaged first by taking a single-frame snapshot. Dynein was imaged every 1 s for 5 min. At the end, microtubules were imaged again by taking a snapshot to assess stage drift. Videos showing significant drift were not analyzed. Each sample was imaged no longer than 15 min.

To assemble human dynein–dynactin–activating adaptor complexes, purified dynein (10–20 nM concentration), dynactin, and the activating adaptor were mixed at 1:2:10 molar ratio and incubated on ice for 10 min. These dynein–dynactin–activating adaptor complexes were then incubated with Lis1 or modified TEV buffer (to buffer match for experiments without Lis1) for 10 min on ice. The mixtures of dynein, dynactin, activating adaptor or dynein alone and Lis1 were then flowed into the flow chamber assembled with taxol-stabilized microtubules. The final imaging buffer contained the assay buffer with 20 μM taxol, 1 mg/mL casein, 7.5 mM potassium chloride, 71.5 mM β-mercaptoethanol, an oxygen

scavenger system, and 2.5 mM Mg-ATP. The final imaging buffer also contained 60 mM potassium acetate for experiments using Hook3 as the activating adaptor and 30 mM potassium acetate for experiments with BicD2. The final concentration of dynein in the flow chamber was 0.5–1. The final concentration of Lis1 was between 12 nM and 300 nM (as indicated in the main text). Imaging was performed as above except with images taken every 300 ms for 3 min.

### Single-molecule microtubule binding assay analysis

Intensity profiles of yeast dynein spots from a single-frame snapshot were generated over a 5-pixel-wide line drawn perpendicular to the long axis of microtubules in ImageJ. Intensity peaks at least twofold higher than the neighboring background intensity were counted as dynein bound to microtubules. Bright aggregates that were fivefold brighter than the neighboring intensity peaks were not counted. The average binding density was calculated as the total number of dynein spots divided by the total microtubule length in each snapshot. Normalized binding density was calculated by dividing the average binding density of dynein without Lis1 collected on the same coverslip (see above). Data plotting and statistical analyses were performed in Prism8 (GraphPad).

### Single-molecule motility assay analysis

Kymographs were generated from motility movies and dynein velocity was calculated from kymographs using ImageJ macros as described (*Roberts et al., 2014*). Only runs that were longer than four frames (4 s) were included in the analysis. Bright aggregates, which were less than 5% of the population, were excluded from the analysis. Data plotting and statistical analyses were performed in Prism8 (GraphPad).

### Microtubule co-pelleting assay

Unlabeled taxol-stabilized MTs were polymerized as above, and free tubulin was removed by centrifugation through a 60% glycerol gradient in BRB80 (80 mM PIPES-KOH pH 6.8, 1 mM magnesium chloride, 1 mM EGTA, 1 mM DTT, 20 µM taxol) for 15 min at $100,000 \times g$ and 37°C. The MT pellet was resuspended in DLB supplemented with 20 µM taxol. MTs (0–600 nM tubulin) were incubated with 100 nM Lis1 for 10 min before being pelleted for 15 min at $100,000 \times g$ and 25°C. The supernatant was analyzed via SDS-PAGE, and depletion was determined using densitometry in ImageJ. Binding curves were fit in Prism8 (GraphPad) with a nonlinear regression for one-site binding with Bmax set to 1. Three technical replicates were collected per condition.

## Acknowledgements

We thank the Nikon Imaging Center at UC San Diego and Dr. Eric Griffis for advice on imaging and analysis and the Cryo-EM Facility at UC San Diego. We also thank our funding sources: JMR is a Merck Fellow of the Damon Runyon Cancer Research Foundation, DRG-2370-19; JPG was funded by the Molecular Biophysics Training Grant, NIH Grant T32 GM008326; EPK was funded by a Jane Coffin Childs Postdoctoral Fellowship; AL's lab by NIH R01 GM107214; and SRP's lab by the Howard Hughes Medical Institute and NIH R35 GM141825. We also thank the reviewers of this work for making this a much better paper.

## Additional information

### Competing interests

Samara L Reck-Peterson: Reviewing editor, eLife. The other authors declare that no competing interests exist.

### Funding

| Funder | Grant reference number | Author |
| --- | --- | --- |
| Howard Hughes Medical Institute | | Samara L Reck-Peterson |

| Funder | Grant reference number | Author |
|---|---|---|
| National Institutes of Health | R35 GM141825 | Samara L Reck-Peterson |
| National Institutes of Health | R01 GM107214 | Andres E Leschziner |
| Damon Runyon Cancer Research Foundation | DRG-2370-19 | Janice M Reimer |
| National Institutes of Health | T32 GM008326 | John P Gillies |
| Jane Coffin Childs Memorial Fund for Medical Research | | Eva P Karasmanis |

The funders had no role in study design, data collection and interpretation, or the decision to submit the work for publication.

## Author contributions

John P Gillies, Eva P Karasmanis, Conceptualization, Formal analysis, Investigation, Methodology, Validation, Visualization, Writing – original draft, Writing – review and editing; Janice M Reimer, Conceptualization, Investigation, Methodology, Validation, Visualization, Writing – original draft, Writing – review and editing; Indrajit Lahiri, Formal analysis, Investigation, Methodology, Validation, Visualization; Zaw Min Htet, Conceptualization, Investigation, Methodology, Validation; Andres E Leschziner, Conceptualization, Investigation, Project administration, Supervision, Writing – original draft, Writing – review and editing; Samara L Reck-Peterson, Conceptualization, Funding acquisition, Project administration, Supervision, Writing – original draft, Writing – review and editing

## Author ORCIDs

John P Gillies ⓘ http://orcid.org/0000-0001-9659-5579
Janice M Reimer ⓘ http://orcid.org/0000-0002-2664-5523
Eva P Karasmanis ⓘ http://orcid.org/0000-0002-0139-0210
Andres E Leschziner ⓘ http://orcid.org/0000-0002-7732-7023
Samara L Reck-Peterson ⓘ http://orcid.org/0000-0002-1553-465X

## Decision letter and Author response

Decision letter https://doi.org/10.7554/eLife.71229.sa1
Author response https://doi.org/10.7554/eLife.71229.sa2

# Additional files

## Supplementary files

• Supplementary file 1. *S. cerevisiae* strains used in this study. A table of the *S. cerevisiae* strains used in this study. *DHA* and *SNAP* refer to the HaloTag (Promega) and SNAP-tag (NEB), respectively. TEV indicates a Tev protease cleavage site. $P_{GAL1}$ denotes the galactose promoter, which was used for inducing strong expression of Lis1 and dynein motor domain constructs. Amino acid spacers are indicated by g (glycine) and gs (glycine-serine).

• Supplementary file 2. Cryo-EM data collection parameters and model refinement statistics. Microscopy and model refinement information for dynein$^{E2488Q}$– (Lis1)$_2$ dataset and structure.

• Transparent reporting form

## Data availability

The Cryo-EM map and model of the dynein-Lis1 complex have been deposited in the EMDB (23829) and the PDB (7MGM).

The following datasets were generated:

| Author(s) | Year | Dataset title | Dataset URL | Database and Identifier |
|---|---|---|---|---|
| Lahiri I, Reimer JM, Leschziner AE | 2022 | Structure of yeast cytoplasmic dynein with AAA3 Walker B mutation bound to Lis1 | https://www.rcsb.org/structure/7MGM | RCSB Protein Data Bank, 7MGM |
| Gillies JP, Reimer JM, Karasmanis EP, Lahiri I, Htet ZM, Leschziner AE, Reck-Peterson SL | 2022 | Structure of yeast cytoplasmic dynein with AAA3 Walker B mutation bound to Lis1 | https://www.ebi.ac.uk/emdb/EMD-23829 | Electron Microscopy Data Bank, EMD-23829 |

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
