## [Editor Report]

This manuscript reports the first high-resolution (3.1Å) structure of a dynein–Lis1 complex. Guided by their cryo-EM structure, the authors make mutations to show that the two Lis1 binding sites (ring and stalk) on the dynein motor are important for both dynein's in vivo function in *S. cerevisiae* and for the formation of human dynein/dynactin complexes.

---

## [Decision Letter]

**Decision letter after peer review:**

Thank you for submitting your article "Structural Basis for Cytoplasmic Dynein-1 Regulation by Lis1" for consideration by *eLife*. Your article has been reviewed by 3 peer reviewers, and the evaluation has been overseen by a Reviewing Editor and Anna Akhmanova as the Senior Editor. The following individuals involved in review of your submission have agreed to reveal their identity: Xin Xiang (Reviewer #1); Deanna Smith (Reviewer #2); Gira Bhabha (Reviewer #3).

The reviewers have discussed their reviews with one another and agreed that this first high resolution structure of a Lis1 bound to dynein is an important piece of work. The Reviewing Editor has drafted this to help you prepare a revised submission.

Essential revisions:

1) The reviewers agreed that your model for the role of Lis1 in controlling affinity for microtubules and determining plus end localization needs more discussion of other papers in the field. Please see reviewer 1 for details.

2) If possible the reviewers ask you to conduct some experiments at high salt under conditions where Lis1 is reported to affect yeast dynein processivity. Please see reviewer 1 for details.

3) The reviewers had some difficulty following the species of LIS1 and dynein, the specific constructs, and the mutations used for each experiment (and also to some extent in the discussion) – can you try to standardize names of constructs and proteins. For example: Reviewer 2 main point, Reviewer 1 point 8, Reviewer 3 point 1, Reviewer 1 point 5.

More detailed questions and suggestions from the reviewers can be found in the full reviews below.

*Reviewer #1 (Recommendations for the authors):*

In this manuscript, the authors reported the first high-resolution (3.1Å) structure of the dynein-Lis1 complex using yeast proteins including the dynein motor domain and LIS1 dimer. This high-resolution structure allows the authors to reveal new details of interaction interfaces, specifically the LIS1-binding site on AAA5, the LIS1-binding site on the stalk as well as the contact site between the two LIS1 propellers (one binds the ring and the other binds stalk). The authors made specific mutations (in LIS1 or dynein) to investigate the significance of these interactions. The functional study was done by using both in vitro (binding curve and effect on in vitro velocity) and in vivo assays (nuclear positioning and dynein localization in yeast). Finally, they also made mutations in human dynein that affect LIS1 binding to the same sites (ring and stalk) and investigated the functional significance of the involved amino acids in cargo-adapter- and dynactin-mediated dynein activation/motility assay. Their results suggest that these sites are important for the formation of active human dynein complexes. Overall, the authors have done a very impressive and beautiful series of experiments. The potential impact of such a high-resolution structure is obvious and their functional analysis has also generated great insights. The data are in general of high quality and nicely presented, and the paper is very well written. However, their model is problematic. It is mainly based on the idea that LIS1 enhances the dynein-microtubule interaction to keep dynein at the microtubule plus end (they think that this tight microtubule binding at the plus end would in turn enhance the formation of the active complex containing dynein, dynactin and cargo adapters). This idea is not consistent with published data (see my point #1 below). Thus, the model figure as well as some discussion points will need to be revised or replaced accordingly. In addition, in vitro assays were done to test whether the mutations affect Pac1/LIS1's ability to induce a tight dynein-microtubule binding or an inhibition of dynein motility, while Marzo, et al., 2020 NCB showed co-migration of dynein and Pac1/LIS1 as well as Pac1/LIS1's ability in enhancing dynein processivity if salt concentration is increased for the in vitro motility assay. I wonder if the authors could try to use higher salt concentrations in the in vitro motility assay, which seems worthwhile because the enhancement of dynein processivity has been linked to the ability of LIS1 in promoting the open state of dynein (the current LIS1-mechanism model that the authors also emphasized in the Discussion). Below I provide more detailed comments for the authors to consider during the revision.

1. Currently, the model emphasized the idea that the LIS1-enhanced dynein-microtubule interaction enhances the microtubule-plus-end accumulation of dynein, which in turn promotes active complex formation. This is inconsistent with experimental findings: In the budding yeast, the microtubule-binding domain is not required for the plus-end accumulation of dynein (Lammers and Markus 2015). In mammalian cells (as well as filamentous fungi), dynactin is needed for the plus-end accumulation of dynein (Xiang et al., 2000; Zhang et al., 2003, 2008, Lenz et al., 2006; Egan et al., 2012, Splinter et al., 2012., Yao et al., 2012) and this is also true in vitro in reconstituted systems (Duellburg et al., 2014; Baumbach et al., 2017; Jha et al., 2017). Would it be better to have a model figure summarizing the current findings on the separation-of-function mutants?

2. It would be better to mention in the Introduction section that LIS1 has recently been implicated in preventing dynein from adopting the autoinhibited phi structure when you mentioned its role in complex formation.

3. Data were presented to argue for an inconsistency with a "tethering" model in which LIS1 tethers dynein to microtubules non-specifically. This was done to argue against the idea proposed by Marzo et al., 2020 that the Pac1/LIS1-caused speed reduction of dynein is due to the non-specific binding of Pac1/LIS1 to microtubules. However, Marzo et al. showed that Pac1/LIS1 does not need to bind dynein to cause a speed reduction, since dynein complexed with Pac1 or not complexed with Pac1 gets the same kind of speed reduction. This point will need to be considered. In addition, since the LIS15A mutant binds to neither dynein nor microtubules, do you have any previous data suggesting that the mutant protein is folded properly or at least expressed well in vivo? This would help strengthen the argument that these amino acids bind microtubules directly. Also, the dimeric LIS1 (LIS1 WT) does seem to lower the velocity more dramatically (Figure 3C), and this also needs to be considered because the argument that dynein-binding and microtubule-binding cannot possibly use the same site of LIS1 only works when a monomer is considered. Given these problems, and the fact that there is no good model explaining how LIS1 affects the mechanochemistry of dynein (aside from phi opening), it may be easier not to put too much effort into this argument but just focus on the new structural data.

4. The interaction between the two LIS1 propellers is important, which is a significant result. Since Htet et al., 2020 NCB showed that LIS1 monomer is effective in promoting active complex formation in vitro, it seems that monomers at a high concentration can still lead to propeller interaction, and you may want to mention this point in this context. Also, it would be nice to cite an early Aspergillus paper showing that LIS1 dimerization is important in vivo (Ahn and Morris 2001 JBC).

5. In Figure 2 G and other figures where you checked LIS1's ability to lower dynein's affinity for microtubules in the presence of ATP-Vi, did you use wild type or the wB-AAA3 mutant?

6. Figure 6, it is unclear where the cortical dynein dots are. It would be better to show a merge with the bright-field images to show cortical dynein or at least use arrows or circles to mark the positions of the signals in all images. Also, would it be better to show representative images of the new mutants rather than just lis1∆?

7. It would be better to at least briefly mention in the main text how you quantify dynein for the binding curves.

8. The dynein mutant defective in binding AAA5 does show a little lowered level, and the authors can state something like "a slightly reduced expression level was observed". Please include a simple sequence alignment figure to show the positions of the mutated amino acids or those in the contacting sites in human dynein and yeast dynein heavy chains.

9. Lis1S248Q and Lis1F185D, I189D, R494A are capable of inducing tight microtubule binding in the presence of ATP but mutants carrying these mutations show a clear nuclear-positioning phenotype in yeast. This at least argues against the tight MT-binding being the only key effect of LIS1 in vivo. What would be the other key effects? "The weakening of dynein's interactions with microtubules when AAA3 is bound to ATP" does not seem to agree with the genetic data from Aspergillus that the wB-AAA3 mutation allows the LIS1 function to be bypassed. It seems more likely that LIS1's effect on promoting the open dynein state is relevant in vivo as this was shown in both yeast and Aspergillus. For these two mutants, I wonder if the open dynein (phi mutant) would suppress their nuclear-positioning defect. If so, that would be very interesting. Although I don't think these experiments are absolutely needed for the publication of this work, it would indeed be better to at least discuss these individual mutants in light of this current model of LIS1 action (i.e. overcoming the phi state and promoting the open state).

10. LIS1 seems to be requited for both the plus-end accumulation and the cortical interaction of dynein, but the beauty of the Lis1S248Q mutation is that it seems to only affect the dynein-dynactin-Num1 complex formation. You should discuss this. Could the nuclear-positioning defect of this mutant be suppressed by the phi mutation or by the AAA3 Walker B mutation? Would this mutation prevent dynein from leaving the microtubule plus end when the Num1 CC domains are overexpressed (Lammers and Markus 2015)? These experiments were doable in Aspergillus only because LIS1 is not required for the plus-end dynein accumulation, but now you have gotten this wonderful function-separating mutant that can be used in yeast for various future assays!

11. "….reach the cell cortex, the site where active yeast dynein-dynactin complexes interact with the candidate activating adaptor, Num1 (Figure 8A, panel iii) (Heil-Chapdelaine et al., 2000; Lammers and Markus, 2015; Lee et al., 2003; Sheeman et al..)". It is better to add Tang et al. 2012 JCB after this statement because this paper dissected the dynein-cortex interaction in detail. Also, it's better not to call the dynein-dynactin complexes "active" because it is only active after Num1 binding (Lammers and Markus 2015).

12. The structure was solved using the yeast dynein motor domain containing a Walker B mutation in AAA3 (E2488Q). Given the recent genetic data from Aspergillus that the wB-AAA3 mutation bypasses the requirement of LIS1 (Qiu et al., 2021, bioRxiv), some people may think that the newly identified interactions are not important in other systems. Thus, it would be better to mention that the human dynein used for structural work (Htet et al., 2020 NCB) is a wild type with linkers bent, but LIS1 binding at the stalk was still revealed, suggesting that these interactions sites play roles in multiple dynein states.

*Reviewer #2 (Recommendations for the authors):*

It was difficult to keep track of dynein and LIS1 constructs used in the study. It might be better to use PAC1 for experiments with the yeast protein and LIS1 in experiments with human protein. Also, the simple term "dynein" should probably be reserved for the holoenzyme and descriptors used to indicate when other forms of dynein are used (monomeric, mutant etc).

*Reviewer #3 (Recommendations for the authors):*

We enjoyed reading this manuscript, which was very well written and presented. We have a few comments below that we ask the authors to address.

1) We had a hard time figuring out what dynein construct was used for the cryo EM structure. Could the authors make this clear in the methods/table please?

2) As the authors note, nucleotide density is somewhat ambiguous. We noted that this may be the case for AAA4 as well. If it was not attempted already, we suggest signal subtraction, re-centering and local refinement perhaps focused on AAA2-3-4, AAA5-6-1L, AAA6-1-2 and around the two Lis1 binding sites. The resolution of this structure is already very impressive and the field would benefit from the highest resolution dynein-Lis1 structure possible; signal subtraction / local refinement may provide additional resolution which could allow to unambiguously assign nucleotide density and some parts of the interface.

3) Line 262 – the authors mention that "Arg 494 was also mutated to Ala to remove the only hydrogen bond donor/acceptor in the interface". However, it seems that D183 and K178 may form a salt bridge between the two Lis1 b-propellers and contribute to the interface?

4) It seems that R3476 (dynein) and E253 (Lis1) could interact if the E253 rotamer was changed, and indeed these residues were chosen to mutate. Since there is not density defining this rotamer, do the authors think that switching the rotamer to facilitate salt bridging would make sense? Currently R3476 looks like it interacts with D3487, so it's possible it can switch between these two interactions.

5) In the structure statistics table (Supplementary file 2) please include: cc values, sphericity value from 3D FSC

6) Great use of the streptavidin affinity grids for a really tough sample!

7) Throughout the manuscript, it may be useful to name the Lis1 chains in some way so it is easy to follow which b-propeller if being referenced at each instance

8) Figure 6: We found the microscopy assay a bit hard to follow, and would appreciate if it were made a bit more clear. A few things we were unsure of:

– Are the measurements being made in the 20% of cells that are the aberrant binucleate cells?

– Was this all done by live cell imaging? Or were cells fixed and analysed?

– Because of the background, the images in Figure 6C are hard to follow. We were wondering if zooming in further may help. A cortical marker or + end marker would also be helpful but may not be necessary. Maybe the authors could draw an outline of the cells to help the readers to see better, which would be an easy fix.

9) Figure 1E-H is a bit hard to follow with the density and the models. We suggest to highlight interactions using just the models, and show the map quality in the supplement.

10) Figure 2E – do the authors find it surprising that the affinity is not changed more? We would have expected to see more of a change in binding affinities.

11) For the mutants that were made, is it clear that they all fold properly/are stable?

12) Could the authors include statistical analysis for Figure 2C,D, Figure 4B and 5B, to test significance as they have done for other assays?

---

## [Author Response]

Essential revisions:1) The reviewers agreed that your model for the role of Lis1 in controlling affinity for microtubules and determining plus end localization needs more discussion of other papers in the field. Please see reviewer 1 for details.

Thank you for this comment, which prompted us to perform a new experiment, revise our model figure, and add to our discussion.

The new experiment we added (new Figure 7) suggests that there is an additional role for Lis1 downstream of disrupting autoinhibited Phi dynein. Our data indicates that this role precedes cortical localization of dynein. Our data does not allow us to determine whether this additional role involves the stabilization of tight microtubule binding by Lis1.

Specifically, we made mutations that disrupt dynein’s Phi conformation (Marzo et al. 2020) and compared the phenotype of this mutant with a mutant that is compromised for Lis1 binding to dynein at site_ring_ (where Lis1 induces tight microtubule binding in vitro), and to a double mutant containing both the Phi disrupting mutation and the Lis1 site_ring_ dynein binding site mutation. We then quantified dynein localization to the yeast cell cortex, where complex assembly of dynein and dynactin with the putative activating adaptor Num1 occurs. Our analysis shows that (1) the Phi disrupting dynein mutant causes more dynein to reach the cortex compared to WT dynein and (2) combining this mutant with the Lis1 site_ring_ dynein binding mutant leads to a decrease in dynein reaching the cortex relative to the Phi disrupting mutant alone. This decrease in cortical dynein relative to the Phi disrupting dynein mutant suggests an additional role for Lis1 in localizing dynein to the cortex beyond disrupting dynein’s autoinhibited conformation. In our discussion we consider possible roles, including promoting additional conformation changes in dynein. This experiment, combined with other experiments in the field, have prompted us to revise and simplify our model figure (new Figure 9).

2) If possible the reviewers ask you to conduct some experiments at high salt under conditions where Lis1 is reported to affect yeast dynein processivity. Please see reviewer 1 for details.

Our main goal with the in vitro assays using yeast proteins was to test and validate the new interactions we identified in our structure of a dynein-Lis1 complex (Figure 1). We wanted well established assays with clear readouts for defects in the interaction between dynein and Lis1. Two of these readouts are the Lis1-dependent changes in dynein velocity and microtubule binding in vitro. Importantly, we were not using these assays to test how disrupting interactions based on our structure affected the physiological function of Lis1 (we used the in vivo assays for that); rather, we were using the assays as tools to test the structure. Given that, we felt that repeating the experiments using different salt conditions would not add to our structural insights and decided against expanding the experiments.

3) The reviewers had some difficulty following the species of LIS1 and dynein, the specific constructs, and the mutations used for each experiment (and also to some extent in the discussion) – can you try to standardize names of constructs and proteins. For example: Reviewer 2 main point, Reviewer 1 point 8, Reviewer 3 point 1, Reviewer 1 point 5.

We agree. In thinking about how to best keep the manuscript easy to read, we decided to clearly indicate in each section of the paper if we are using yeast or human proteins. For example, within each section of the results only one species is used, and we indicate this at the beginning of each section. In the introduction and discussion, we have added language referring to the specific organisms, where appropriate. We have made these changes throughout the manuscript. However, if the reviewers still find the manuscript unclear, we would be happy to consider the other possibilities suggested by them.

For the AAA3 Walker B mutant construct of dynein used for the structural work, we have changed the name to dynein^E2488Q^.

More detailed questions and suggestions from the reviewers can be found in the full reviews below.

In addition to adding a new Figure 7 and revising our model figure (now new Figure 9), we have made several other major changes and additions to the manuscript, which include:

1) A new Figure 2 and movie 2, which highlight our structure of yeast Lis1 and the interaction sites for dynein and Lis1.

2) We removed what was originally Figure 3 as we have decided that our experiments on microtubule binding distracted from the main message of our paper and that additional experiments are warranted that would go beyond the scope of the current study.

Reviewer #1 (Recommendations for the authors):In this manuscript, the authors reported the first high-resolution (3.1Å) structure of the dynein-Lis1 complex using yeast proteins including the dynein motor domain and LIS1 dimer. This high-resolution structure allows the authors to reveal new details of interaction interfaces, specifically the LIS1-binding site on AAA5, the LIS1-binding site on the stalk as well as the contact site between the two LIS1 propellers (one binds the ring and the other binds stalk). The authors made specific mutations (in LIS1 or dynein) to investigate the significance of these interactions. The functional study was done by using both in vitro (binding curve and effect on in vitro velocity) and in vivo assays (nuclear positioning and dynein localization in yeast). Finally, they also made mutations in human dynein that affect LIS1 binding to the same sites (ring and stalk) and investigated the functional significance of the involved amino acids in cargo-adapter- and dynactin-mediated dynein activation/motility assay. Their results suggest that these sites are important for the formation of active human dynein complexes. Overall, the authors have done a very impressive and beautiful series of experiments. The potential impact of such a high-resolution structure is obvious and their functional analysis has also generated great insights. The data are in general of high quality and nicely presented, and the paper is very well written. However, their model is problematic. It is mainly based on the idea that LIS1 enhances the dynein-microtubule interaction to keep dynein at the microtubule plus end (they think that this tight microtubule binding at the plus end would in turn enhance the formation of the active complex containing dynein, dynactin and cargo adapters). This idea is not consistent with published data (see my point #1 below). Thus, the model figure as well as some discussion points will need to be revised or replaced accordingly. In addition, in vitro assays were done to test whether the mutations affect Pac1/LIS1's ability to induce a tight dynein-microtubule binding or an inhibition of dynein motility, while Marzo, et al., 2020 NCB showed co-migration of dynein and Pac1/LIS1 as well as Pac1/LIS1's ability in enhancing dynein processivity if salt concentration is increased for the in vitro motility assay. I wonder if the authors could try to use higher salt concentrations in the in vitro motility assay, which seems worthwhile because the enhancement of dynein processivity has been linked to the ability of LIS1 in promoting the open state of dynein (the current LIS1-mechanism model that the authors also emphasized in the Discussion). Below I provide more detailed comments for the authors to consider during the revision.1. Currently, the model emphasized the idea that the LIS1-enhanced dynein-microtubule interaction enhances the microtubule-plus-end accumulation of dynein, which in turn promotes active complex formation. This is inconsistent with experimental findings: In the budding yeast, the microtubule-binding domain is not required for the plus-end accumulation of dynein (Lammers and Markus 2015). In mammalian cells (as well as filamentous fungi), dynactin is needed for the plus-end accumulation of dynein (Xiang et al., 2000; Zhang et al., 2003, 2008, Lenz et al., 2006; Egan et al., 2012, Splinter et al., 2012., Yao et al., 2012) and this is also true in vitro in reconstituted systems (Duellburg et al., 2014; Baumbach et al., 2017; Jha et al., 2017). Would it be better to have a model figure summarizing the current findings on the separation-of-function mutants?

Please see our response to essential revision #1.

Would it be better to have a model figure summarizing the current findings on the separation-of-function mutants?

We now emphasize that this is a separation-of-function mutation in Lis1 in the results with the following sentence:

“We note that the site_stalk_ mutant is the only Lis1 mutant with no defect in dynein’s localization to microtubule plus ends, thus this separation-of-function mutant could be a useful tool to further understand the mechanism of dynein regulation by Lis1.”

2. It would be better to mention in the Introduction section that LIS1 has recently been implicated in preventing dynein from adopting the autoinhibited phi structure when you mentioned its role in complex formation.

We thank the reviewer for this suggestion. We have revised our introduction accordingly to say:

“Recently, Lis1 was shown to play a role in forming active dynein complexes, most likely by disrupting dynein’s autoinhibited Phi conformation (Elshenawy et al., 2020; Htet et al., 2020; Marzo et al., 2020; Qiu et al., 2019).”

3. Data were presented to argue for an inconsistency with a "tethering" model in which LIS1 tethers dynein to microtubules non-specifically. This was done to argue against the idea proposed by Marzo et al., 2020 that the Pac1/LIS1-caused speed reduction of dynein is due to the non-specific binding of Pac1/LIS1 to microtubules. However, Marzo et al. showed that Pac1/LIS1 does not need to bind dynein to cause a speed reduction, since dynein complexed with Pac1 or not complexed with Pac1 gets the same kind of speed reduction. This point will need to be considered. In addition, since the LIS15A mutant binds to neither dynein nor microtubules, do you have any previous data suggesting that the mutant protein is folded properly or at least expressed well in vivo? This would help strengthen the argument that these amino acids bind microtubules directly. Also, the dimeric LIS1 (LIS1 WT) does seem to lower the velocity more dramatically (Figure 3C), and this also needs to be considered because the argument that dynein-binding and microtubule-binding cannot possibly use the same site of LIS1 only works when a monomer is considered. Given these problems, and the fact that there is no good model explaining how LIS1 affects the mechanochemistry of dynein (aside from phi opening), it may be easier not to put too much effort into this argument but just focus on the new structural data.

We have decided to remove our experiments on microtubule binding from the manuscript as we feel that they distracted from the main message of our paper and that additional experiments are warranted that go beyond the scope of the current study. However, we still address each point brought up by the reviewer below.

The data you are referring to comes from Marzo et al. Figure 5D. In this experiment, the authors examine full length dynein velocity and run length at a single concentration of Pac1/Lis1 (25 nM) in 150 mM KCl.

From our work, the apparent Kd for the dynein/Pac1 interaction in 50 mM KAcetate is greater than 100 nM. Thus, at 25 nM Pac1 (1/4 x Kd) the occupancy of Pac1 on dynein should be less than 30%. This is likely to be even lower at the 150 mM KCl used in the Marzo et al. work. Given this, we would expect Pac1 to be coming on and off dynein, with a small fraction of complexes formed at any given time.

We have found that Pac1 decreases dynein velocity in a dose-dependent manner (Huang and Roberts et al., 2012). Our current hypothesis is that Pac1/Lis1’s major function is to relieve the inhibition of dynein when it is in a Phi conformation, something that may happen at low (catalytic) concentrations of Pac1 (at least in vitro, the Phi conformation tends not to reform once it has been disrupted). Once dynein has assumed a parallel active conformation, any binding of Pac1 to dynein at site_ring_ would decrease dynein’s velocity by biasing dynein towards a tight microtubule binding conformation (the “open” ring state). This function, however, requires stoichiometric binding of Pac1 to dynein. Therefore, we suspect that Marzo and colleagues might have observed a reduction in dynein velocity if a range of Pac1 concentrations had been tried in their assay.

4. The interaction between the two LIS1 propellers is important, which is a significant result. Since Htet et al., 2020 NCB showed that LIS1 monomer is effective in promoting active complex formation in vitro, it seems that monomers at a high concentration can still lead to propeller interaction, and you may want to mention this point in this context.

Please see our response to point 3 above.

Also, it would be nice to cite an early Aspergillus paper showing that LIS1 dimerization is important in vivo (Ahn and Morris 2001 JBC).

Thank you for the suggestion. We have added the following to the introduction:

“Structurally, Lis1 is a dimer of two ß-propellers (Kim et al., 2004; Tarricone et al., 2004), and in vivo Lis1 dimerization is required for its function (Ahn and Morris, 2001; Huang et al., 2012).”

5. In Figure 2 G and other figures where you checked LIS1's ability to lower dynein's affinity for microtubules in the presence of ATP-Vi, did you use wild type or the wB-AAA3 mutant?

We used wild type dynein. We have revised the figure legends to clarify this.

6. Figure 6, it is unclear where the cortical dynein dots are. It would be better to show a merge with the bright-field images to show cortical dynein or at least use arrows or circles to mark the positions of the signals in all images. Also, would it be better to show representative images of the new mutants rather than just lis1∆?

We have added the cell outlines in Figure 6C and created a new Figure 6—figure supplement 1, which contains a gallery of the mutants. We added outlines to new Figure 7 as well.

7. It would be better to at least briefly mention in the main text how you quantify dynein for the binding curves.

We have done this.

8. The dynein mutant defective in binding AAA5 does show a little lowered level, and the authors can state something like "a slightly reduced expression level was observed". Please include a simple sequence alignment figure to show the positions of the mutated amino acids or those in the contacting sites in human dynein and yeast dynein heavy chains.

We have repeated the dynein GFP pull downs three times and the levels of our dynein^ΔN3475, ΔR3476^ are similar to those of WT dynein. The quantification below shows the band intensity (in arbitrary units) of dynein-3xGFP WT, dynein^ΔN3475, ΔR3476^-3xGFP and dyneinΔ +/- s.e.m from three independent GFP pull down experiments.

**Author response image 1. sa2fig1:** Quantification of Dynein-3XGFP band intensity +/- s. e.m.

We previously published a sequence alignment around the putative site_stalk_ (DeSantis et al., 2017 Figure 5A).

9. Lis1S248Q and Lis1F185D, I189D, R494A are capable of inducing tight microtubule binding in the presence of ATP but mutants carrying these mutations show a clear nuclear-positioning phenotype in yeast. This at least argues against the tight MT-binding being the only key effect of LIS1 in vivo. What would be the other key effects? "The weakening of dynein's interactions with microtubules when AAA3 is bound to ATP" does not seem to agree with the genetic data from Aspergillus that the wB-AAA3 mutation allows the LIS1 function to be bypassed. It seems more likely that LIS1's effect on promoting the open dynein state is relevant in vivo as this was shown in both yeast and Aspergillus. For these two mutants, I wonder if the open dynein (phi mutant) would suppress their nuclear-positioning defect. If so, that would be very interesting. Although I don't think these experiments are absolutely needed for the publication of this work, it would indeed be better to at least discuss these individual mutants in light of this current model of LIS1 action (i.e. overcoming the phi state and promoting the open state).

Please see our response to essential revisions point #1.

10. LIS1 seems to be requited for both the plus-end accumulation and the cortical interaction of dynein, but the beauty of the Lis1S248Q mutation is that it seems to only affect the dynein-dynactin-Num1 complex formation. You should discuss this. Could the nuclear-positioning defect of this mutant be suppressed by the phi mutation or by the AAA3 Walker B mutation? Would this mutation prevent dynein from leaving the microtubule plus end when the Num1 CC domains are overexpressed (Lammers and Markus 2015)? These experiments were doable in Aspergillus only because LIS1 is not required for the plus-end dynein accumulation, but now you have gotten this wonderful function-separating mutant that can be used in yeast for various future assays!

We agree that these are interesting questions and exciting new directions that we would love to follow up on in the future. We have highlighted in the manuscript that this is a function separating mutation.

11. "….reach the cell cortex, the site where active yeast dynein-dynactin complexes interact with the candidate activating adaptor, Num1 (Figure 8A, panel iii) (Heil-Chapdelaine et al., 2000; Lammers and Markus, 2015; Lee et al., 2003; Sheeman et al..)". It is better to add Tang et al. 2012 JCB after this statement because this paper dissected the dynein-cortex interaction in detail. Also, it's better not to call the dynein-dynactin complexes "active" because it is only active after Num1 binding (Lammers and Markus 2015).

We have modified these parts of the paper extensively.

We intended to only call dynein-dynactin-activating adaptor complexes “active”. When we use the term “activated dynein-dynactin complexes” to us this is synonymous with saying dyneindynactin-activating adaptor complexes. We have explicitly stated this in the text now to prevent any confusion:

“Active moving dynein complexes contain the dynactin complex and a coiled-coil-containing activating adaptor, referred to here as “active dynein–dynactin” or “dynein–dynactin–activating adaptor complexes” (Mckenney et al., 2014; Schlager et al., 2014; Trokter et al., 2012) (Figure 1A).”

12. The structure was solved using the yeast dynein motor domain containing a Walker B mutation in AAA3 (E2488Q). Given the recent genetic data from Aspergillus that the wB-AAA3 mutation bypasses the requirement of LIS1 (Qiu et al., 2021, bioRxiv), some people may think that the newly identified interactions are not important in other systems. Thus, it would be better to mention that the human dynein used for structural work (Htet et al., 2020 NCB) is a wild type with linkers bent, but LIS1 binding at the stalk was still revealed, suggesting that these interactions sites play roles in multiple dynein states.

We have pointed out our 3D class averages of the human dynein plus Lis1 used wild type protein (Htet et al., 2020) in the Discussion:

“Our previous work with wild type human dynein and Lis1 showed that Lis1 binds to dynein at similar sites.”

Reviewer #2 (Recommendations for the authors):It was difficult to keep track of dynein and LIS1 constructs used in the study. It might be better to use PAC1 for experiments with the yeast protein and LIS1 in experiments with human protein. Also, the simple term "dynein" should probably be reserved for the holoenzyme and descriptors used to indicate when other forms of dynein are used (monomeric, mutant etc).

We agree that we needed to improve our nomenclature throughout. We have made modifications throughout the text that we hope increase clarity. Please see our response to essential revisions #3.

Reviewer #3 (Recommendations for the authors):We enjoyed reading this manuscript, which was very well written and presented. We have a few comments below that we ask the authors to address.1) We had a hard time figuring out what dynein construct was used for the cryo EM structure. Could the authors make this clear in the methods/table please?

Our apologies for the confusion. We have made the following additions to the manuscript: (1) The column header in supplementary file 2 was changed from “Dynein AAA3 Walker B bound to Lis1” to “Dynein AAA3^E2488Q^ bound to Lis1.” (2) We have updated the paper throughout to specify each dynein construct using amino acid changes.

2) As the authors note, nucleotide density is somewhat ambiguous. We noted that this may be the case for AAA4 as well. If it was not attempted already, we suggest signal subtraction, re-centering and local refinement perhaps focused on AAA2-3-4, AAA5-6-1L, AAA6-1-2 and around the two Lis1 binding sites. The resolution of this structure is already very impressive and the field would benefit from the highest resolution dynein-Lis1 structure possible; signal subtraction / local refinement may provide additional resolution which could allow to unambiguously assign nucleotide density and some parts of the interface.

Thank you for this suggestion. We generated signal-subtracted reconstructions for AAA5L-6-1-2S, AAA2-3-4-5S, AAA2-3-4-5S+Lis1_ring_+Lis1_stalk_, and Lis1_ring_ + Lis1_stalk_ alone. Author response image 2; Author response image 3 present our results. Author response image 2 shows local resolution maps for the signal-subtracted reconstructions along with that for the corresponding portion of the original structure. Author response image 3 shows close-ups from the only map where we saw an improvement in the resolution (AAA2-3-4-5S+Lis1_ring_+Lis1_stalk_).

**Author response image 2. sa2fig2:** Signal-subtracted reconstructions for dynein^E2488Q^-2x(Lis1). Local resolution is shown for signal subtracted reconstructions of (A) AAA5L-6-1-2S, (B) AAA2-3-4-5S+Lis1_ring_+Lis1_stalk_, (C) AAA2-3-4-5S and (D) Lis1_ring_ + Lis1_stalk_ alone. The mask used for signal subtraction is shown on the original map. For comparison, the original portion of the map is displayed next to the resulting signal subtracted reconstruction.

**Author response image 3. sa2fig3:** Examples of map density for signal-subtracted reconstruction of AAA2-3-4-5S+Lis1_ring_+Lis1_stalk_. (A) The map quality for ATP at AAA3 is only nominally improved in the signal-subtracted reconstruction and remains unclear whether ATP or ADP-Vi is bound in the active site. (B) The signal-subtracted map shows additional density connected to R3512, which could correspond to a flexible phosphate group or vanadate ion. (C) Signal subtraction did not improve the resolution of the newly identified AAA5-Lis1_ring_ interaction site. There was some resolution improvement for sidechains throughout the map (example, Lis1 K280).

These are the main observations from these reprocessed maps:

1) The signal-subtracted map for Lis1_ring_ + Lis1_stalk_ alone is significantly worse than the original one, likely due to the relatively low molecular weight of the two b-propellers.

2) The signal-subtracted map for AAA5L-6-1-2S has only a nominal increase in resolution.

3) Similarly, the signal-subtracted map for AAA2-3-4-5S shows a nominal increase in resolution.

4) The largest, albeit still relatively modest, improvement in resolution was for the signal subtracted map of AAA2-3-4-5S+Lis1_ring_+Lis1_stalk_.

5) Regardless of the improvement, or lack thereof, in the resolution of the signal-subtracted maps, we did not detect any major structural differences relative to the full map.

6) The nucleotide states, in terms of distinguishing between ATP and ADP.Vi, remain ambiguous.

7) AAA4 in AAA2-3-4-5S+Lis1_ring_+Lis1_stalk_ does show an interesting new feature. If we lower the threshold of the map (see middle panel, right, in Author response image 3), we observe a density connected to R3512. It is possible that this density corresponds to a flexible phosphate or vanadate, which would be adjacent to the β phosphate of the ATP found at AAA4. Although intriguing, this is purely speculative at this point.

8) Despite the improved resolution of the AAA2-3-4-5S+Lis1_ring_+Lis1_stalk_ map, the density for the AAA5 loop that is part of the binding site for Lis1_ring_ did not improve (close-up in Author response image 3).

9) We will deposit the signal-subtracted map for AAA2-3-4-5S+Lis1_ring_+Lis1_stalk_. Even if this map did not change our model for the complex, its higher resolution and novel feature at AAA4 would make it interesting to the community.

3) Line 262 – the authors mention that "Arg 494 was also mutated to Ala to remove the only hydrogen bond donor/acceptor in the interface". However, it seems that D183 and K178 may form a salt bridge between the two Lis1 b-propellers and contribute to the interface?

We were mistaken to say that R494 forms the only hydrogen bond in the interface as D183 and K178 do have the potential to form a salt bridge. K178 was not selected for mutation because it is in close proximity to Site_stalk_ and we did not want to risk interfering with this site. We have changed the manuscript accordingly.

4) It seems that R3476 (dynein) and E253 (Lis1) could interact if the E253 rotamer was changed, and indeed these residues were chosen to mutate. Since there is not density defining this rotamer, do the authors think that switching the rotamer to facilitate salt bridging would make sense? Currently R3476 looks like it interacts with D3487, so it's possible it can switch between these two interactions.

The most probable rotamer positions for Lis1 E253 orient the sidechain so that it is either facing upwards or rotated towards dynein R3476, which is why it was selected for mutation. However, real space refinement of the model consistently places the E253 side chain so that it is facing towards Lis1. As there is no density to guide the placement the sidechain, we chose not to manually alter the refined model.

5) In the structure statistics table (Supplementary file 2) please include: cc values, sphericity value from 3D FSC

The structure statistics table has been updated to include these values.

6) Great use of the streptavidin affinity grids for a really tough sample!

Thank you!

7) Throughout the manuscript, it may be useful to name the Lis1 chains in some way so it is easy to follow which b-propeller if being referenced at each instance

We discuss the interaction between Lis1 and dynein in both broad and specific terms. When the specific location of Lis1 binding is important to the reader, we use the terms site_ring_ and site_stalk_ to denote which Lis1 is being referred to. We feel that it would be redundant to label both sitering/sitestalk and Lis1ring/Lis1stalk.

8) Figure 6: We found the microscopy assay a bit hard to follow, and would appreciate if it were made a bit more clear. A few things we were unsure of:– Are the measurements being made in the 20% of cells that are the aberrant binucleate cells?

The yeast strains used for Figure 6, Figures 2C, 4B and 5B are genetically modified at their endogenous loci to either delete yeast Lis1or insert the desired mutations. Because dynein is a non-essential gene in *S. cerevisiae*, in our multinucleated assays, only ~20% of the cells are multinucleate in the absence of dynein or Lis1. However, 100% of the cells contain the outlined mutation.

– Was this all done by live cell imaging? Or were cells fixed and analysed?

We apologize for not including this information before. We have added a new section in our Materials and methods clearly describing the methodology for Figure 6 and 7.

The cells for multinucleated assays (Figures 2C, 4B, 5B, and 7A) are fixed and stained with DAPI.

In the dynein localization experiments (Figure 6 and 7B-C), we use live-cell imaging. Each marker (3X-GFP-Dyn1, SPB-mcherry and TUB1-CFP) is tagged at their endogenous locus. We mount the cells on agarose pads and perform live-cell imaging using a spinning disk confocal microscope.

– Because of the background, the images in Figure 6C are hard to follow. We were wondering if zooming in further may help. A cortical marker or + end marker would also be helpful but may not be necessary. Maybe the authors could draw an outline of the cells to help the readers to see better, which would be an easy fix.

We have added an outline to the cells, thank you for the suggestion!

9) Figure 1E-H is a bit hard to follow with the density and the models. We suggest to highlight interactions using just the models, and show the map quality in the supplement.

Thank you for the suggestion. Figure 1 was redesigned to remove the maps and showcase the three major interaction sites using just the models. We have also added a new Figure 2 that focuses on Lis1 and highlights the surfaces involved in its different interactions. This new figure allowed us to shift some of the details about the dynein-Lis1 and Lis1-Lis1 interactions away from Figure 1, keeping that first figure cleaner and broader.

10) Figure 2E – do the authors find it surprising that the affinity is not changed more? We would have expected to see more of a change in binding affinities.

The site_ring_ interface is comprised of two different interactions between Lis1 and dynein: Lis1 with the AAA4 helix and Lis1 with the AAA5 loop (Figure 1, D). We previously showed that the AAA4 helix – Lis1 interaction is important for Lis1 binding (Toropova et al., 2014). The mutations tested here aimed to disrupt the new Lis1 interaction with AAA5. This contact only accounts for ~20% of the buried surface area between the two proteins (as estimated using PDBePISA), with the rest of the contact being made between Lis1 and AAA4. Given the importance of the large Lis1-AAA4 interaction, we would expect Lis1 to still be capable of binding at AAA4, and thus mutations at the Lis1-AAA5 interaction would only cause a partial disruption in binding at site_ring_.

One of the panels in the new Figure 2 shows residues we had previously mutated in either dynein (Huang, Roberts et al., 2013) or Lis1 (Toropova et al., 2014) to disrupt binding of Lis1 at site_ring_. We hope this panel in the new figure would help clarify that the mutations tested in our current work only disrupt part of the site_ring_ interaction.

11) For the mutants that were made, is it clear that they all fold properly/are stable?

We have not directly tested this, but we have determined that expression levels are unchanged in cells. We are also able to purify all mutant proteins and perform in vitro assays with them.

12) Could the authors include statistical analysis for Figure 2C,D, Figure 4B and 5B, to test significance as they have done for other assays?

We have revised our figures and figure legends for figures 2C,D (which is now Figure 3C, D), Figure 4B and 5B to include a statistical analysis of our assays.